# ER calcium depletion as a key driver for impaired ER-to-mitochondria calcium transfer and mitochondrial dysfunction in Wolfram syndrome

Mailis Liiv [1,8], Annika Vaarmann [1,8,9] ✉, Dzhamilja Safiulina [1], Vinay Choubey [1], Ruby Gupta[1], Malle Kuum[1], Lucia Janickova [1,2,3], Zuzana Hodurova[1,3], Michal Cagalinec [1,4], Akbar Zeb[1], Miriam A. Hickey [1], Yi-Long Huang [5], Nana Gogichaishvili[1], Merle Mandel[1], Mario Plaas[1], Eero Vasar [1], Jens Loncke[6], Tim Vervliet[6], Ting-Fen Tsai[5], Geert Bultynck [6], Vladimir Veksler[7] & Allen Kaasik [1,9] ✉

Wolfram syndrome is a rare genetic disease caused by mutations in the WFS1 or CISD2 gene. A primary defect in Wolfram syndrome involves poor ER $Ca^{2+}$ handling, but how this disturbance leads to the disease is not known. The current study, performed in primary neurons, the most affected and disease-relevant cells, involving both Wolfram syndrome genes, explains how the disturbed ER $Ca^{2+}$ handling compromises mitochondrial function and affects neuronal health. Loss of ER $Ca^{2+}$ content and impaired ER-mitochondrial contact sites in the WFS1- or CISD2-deficient neurons is associated with lower $IP_3R$-mediated $Ca^{2+}$ transfer from ER to mitochondria and decreased mitochondrial $Ca^{2+}$ uptake. In turn, reduced mitochondrial $Ca^{2+}$ content inhibits mitochondrial ATP production leading to an increased $NADH/NAD^+$ ratio. The resulting bioenergetic deficit and reductive stress compromise the health of the neurons. Our work also identifies pharmacological targets and compounds that restore $Ca^{2+}$ homeostasis, enhance mitochondrial function and improve neuronal health.

Wolfram syndrome (WS) is a rare genetic disease caused by mutations in the wolframin gene (WFS1), resulting in the more prevalent type I WS, or in the CDGSH iron-sulfur-containing domain 2 gene (CISD2), resulting in the rarer type II WS[1,2]. The primary symptoms of WS (insulin-dependent diabetes mellitus, optic atrophy, diabetes insipidus and hearing loss) can emerge at different ages and change at different rates[3–5]. The current treatment of WS is symptomatic and supportive. Diabetes insipidus is managed by treatment with vasopressin and

[1]Departments of Pharmacology and Physiology, Institute of Biomedicine and Translational Medicine, University of Tartu, Ravila 19, 50411 Tartu, Estonia. [2]Chair of Pharmacology, Faculty of Science and Medicine, University of Fribourg, Ch. du Musée 14, 1700 Fribourg, Switzerland. [3]Department of Cell Pharmacology and Developmental Toxicology, Institute of Experimental Pharmacology and Toxicology, Dúbravská cesta 9, 84104 Bratislava, Slovakia. [4]Department of Cellular Cardiology, Institute of Experimental Endocrinology, Biomedical Research Center and Centre of Excellence for Advanced Materials Application, Slovak Academy of Sciences, Dúbravská cesta 9, 84505 Bratislava, Slovakia. [5]Department of Life Sciences, Institute of Genome Sciences and Center for Healthy Longevity and Aging Sciences, National Yang Ming Chiao Tung University, 155 Li-Nong St., Section 2, Peitou, Taipei 11221, Taiwan. [6]Laboratory of Molecular and Cellular Signaling, Department of Cellular and Molecular Medicine, KU Leuven, O&N1 Herestraat 49, Leuven, Belgium. [7]Laboratory of Signaling and Cardiovascular Pathophysiology, Université Paris-Saclay, Inserm, UMR-S 1180, 91400 Orsay, France. [8]These authors contributed equally: Mailis Liiv, Annika Vaarmann. [9]These authors jointly supervised this work: Annika Vaarmann, Allen Kaasik. ✉e-mail: annika.vaarmann@ut.ee; allen.kaasik@ut.ee

diabetes mellitus by insulin; recent case reports[6,7] suggest a benefit of glucagon-like peptide-1 (GLP-1) receptor agonists. However, so far, there is no treatment available for the neurological manifestations of WS. Relatively slow progress in WS treatment is related to a lack of knowledge of which cellular mechanisms are responsible for neurological symptoms.

Both WS genes encode transmembrane endoplasmic reticulum (ER) proteins[5] and it is widely acknowledged that the primary defect in WS lies also in the ER[8–11]. However, recent discoveries indicate that while the primary defect may indeed originate in the ER, the key factor driving the disease progression might lie within the mitochondria. Our previous research has demonstrated that WFS1 downregulation in cortical neurons disturbed mitochondrial dynamics and suppressed mitochondrial ATP production[11]. Angebault et al.[12] further demonstrated in WS patient fibroblasts that loss of WFS1 reduced the number of mitochondrial contact sites with the ER (MAMs), decreased $Ca^{2+}$ uptake by mitochondria, and decreased mitochondrial respiration. Reduction in MAMs, mitochondrial respiration, and ATP production was then also observed in WS patient-derived neuronal cells[13]. Furthermore, most recent findings by Crouzier et al.[14] suggest that the ER-to-mitochondria $Ca^{2+}$ flux could be corrected with Sigma-1 receptor agonist (which modulates $Ca^{2+}$ signaling through the $IP_3$ receptors), improving mitochondrial physiology in patient fibroblasts and alleviating the behavioral symptoms observed in zebrafish and mouse models of the disease. Notably, mutations in CISD2 are also associated with mitochondrial abnormalities[15–17].

Nevertheless, despite these recent discoveries, several important questions remain unanswered:

- Why is there a reduced flow of $Ca^{2+}$ from the ER to the mitochondria? Is this due to a loss of functional MAMs or a deficiency of $Ca^{2+}$ within the ER?
- How does the diminished influx of $Ca^{2+}$ affect mitochondrial function, and what specific implications does this have for cellular processes?
- Do WFS1 and CISD2 act synergistically to provoke these anomalies, or do they exert separate yet similar effects?

Addressing these questions promises to provide crucial insights into the mechanisms underlying Wolfram syndrome and may suggest new potential therapeutic strategies.

The current study, performed in the most affected and disease-relevant cells (neurons) and cell compartments (axons) and comparing deficiencies of both WS proteins, WFS1 and CISD2, gives answers to these questions. Moreover, our study identifies potential drug targets that may restore $Ca^{2+}$ homeostasis, thereby enhancing mitochondrial function and improving neuronal function.

## Results

### Decreased capacity of SERCA to transport $Ca^{2+}$ from axoplasm to ER

Our first aim was to understand what happens with axonal ER $Ca^{2+}$ store content in WS neurons. We estimated the resting free $Ca^{2+}$ levels in the ER and cytosol of axons (axoplasm) of WFS1- and CISD2-deficient neurons (Supplementary Fig. 1 shows shRNA validation; *Wfs1* shRNA has been validated previously[11]). We employed the ER $Ca^{2+}$ sensor ER-GCaMP6-210, the affinity of which is matched to resting axonal $[Ca^{2+}]_{ER}$, and the cytosolic $Ca^{2+}$ sensor jGCaMP7b, which shows relatively high baseline fluorescence at low resting axoplasmic $[Ca^{2+}]_{cyto}$. It has been observed previously that the GCaMP6 family of fluorophores exhibits dual excitation peaks at 410 and 474 nm, which can be used for ratiometric imaging. By taking the ratio between these wavelengths, the fluorescent signals remain proportional to the $Ca^{2+}$ concentration but independent of probe expression levels (Fig. 1a, b)[18]. We found that lower basal ER $Ca^{2+}$ levels (Fig. 1c) were associated with

elevated resting axoplasmic $Ca^{2+}$ (Fig. 1d). This suggests that ER $Ca^{2+}$ handling is compromised in the axons of WFS1- and CISD2-deficient neurons, leaving more $Ca^{2+}$ in the axoplasm.

This could be caused by a decreased capacity of the ER to sequester $Ca^{2+}$ from the axoplasm, or increased $Ca^{2+}$ release/leak from the ER, or both. Therefore, we first tested whether we could restore the ER $Ca^{2+}$ handling deficit by overexpressing ER $Ca^{2+}$ pump SERCA2. Indeed, overexpression of SERCA2b restored ER $Ca^{2+}$ levels (Fig. 1e) and axoplasmic $Ca^{2+}$ levels (Fig. 1f) back to normal in the axons of WFS1-and CISD2-deficient neurons. Next, we studied the effect of CDN1163, a pharmacological SERCA activator. Both the ER and axoplasmic $Ca^{2+}$ levels were normalized when the WFS1-deficient neurons were treated with the CDN1163 (0.5 μM, a concentration that was derived from a concentration range previously shown to enhance SERCA activity[19,20] and in-house toxicity assay) (Fig. 1g). We also performed additional control experiments to directly monitor variations in ER $Ca^{2+}$ absorption during SERCA overexpression, activation, or inhibition, showing that all these treatments indeed affect SERCA activity (Supplementary Fig. 2A).

We also conducted a reverse experiment where we suppressed SERCA activity by overexpressing phospholamban (PLB, a SERCA negative regulator) and investigated whether the $Ca^{2+}$ homeostasis could be restored by overexpression of WFS1 or CISD2. As shown in Fig. 1h, i, the overexpression of PLB lowered the ER $Ca^{2+}$ levels and augmented axoplasmic $Ca^{2+}$ levels, resembling the $Ca^{2+}$ changes observed in WFS1 or CISD2 knock-down neurons. Moreover, overexpression of either WFS1 or CISD2 counteracted the effect of PLB overexpression on axoplasmic $Ca^{2+}$ homeostasis (Fig. 1i).

To confirm that the decreased ER $Ca^{2+}$ directly reflects SERCA function and is not influenced by other factors such as extracellular $Ca^{2+}$ reuptake, we also measured the $Ca^{2+}$ uptake to the ER in permeabilized cells under a constant $Ca^{2+}$ concentration. The ER $Ca^{2+}$ uptake remained lower in WFS1- and CISD2-deficient neurons, indicating a primary intrinsic deficit in ER $Ca^{2+}$ handling and is not related to external factors (Fig. 1j). We also conducted a single-neuron expression analysis, indicating that the levels of Sarcoplasmic/endoplasmic reticulum calcium ATPase type 2 (SERCA2) were lower in the Wfs1 or Cisd2 knock-down neurons while the expression of ryanodine receptor (RyR) isoform 2 (RyR2) and inositol 1,4,5-trisphosphate ($IP_3$) receptor type 1 ($IP_3R1$) expression levels remained unaffected (Supplementary Fig. 3).

Thus, in WFS1-and CISD2-deficient neurons, SERCA activity is compromised and/or not sufficient to counteract possible ER $Ca^{2+}$ leak. This leads to lowered ER $Ca^{2+}$ store capacity and elevated axoplasmic $Ca^{2+}$ levels.

### Leak through RyR receptors contributes to ER $Ca^{2+}$ loss

Next, we studied whether $Ca^{2+}$ leak through the ER $Ca^{2+}$ release channels, RyR or $IP_3R$ contributed to the ER $Ca^{2+}$ depletion. We first examined the thapsigargin-induced surge of cytosolic $Ca^{2+}$, reflecting mainly the RyR-mediated ER $Ca^{2+}$ leak (Supplementary Fig. 2b). The results show that neurons deficient in WFS1 and CISD2 have significantly less $Ca^{2+}$ available for release in the ER (Fig. 2b).

We then reduced the RyR2 (the predominant ryanodine receptor isoform in the brain[21]) and examined whether this knock-down (Supplementary Fig. 1) could restore axoplasmic and ER $Ca^{2+}$ balance. Figure 2b, c demonstrate that RyR2 knock-down in WFS1-deficient neurons restored $Ca^{2+}$ levels in both ER and axoplasm to those observed in control conditions. A similar effect was observed in WFS1-deficient neurons treated with azumolene, a non-fluorescent analog of dantrolene that inhibits RyR-mediated $Ca^{2+}$ release (Fig. 2d, e) or with a specific inhibitor of RyR-mediated ER $Ca^{2+}$ leak, Rycal S107 (Fig. 2f, g).

We further tested whether suppression of $IP_3R$ induced a similar level of protection. However, as demonstrated in Fig. 2h, i, the knock-

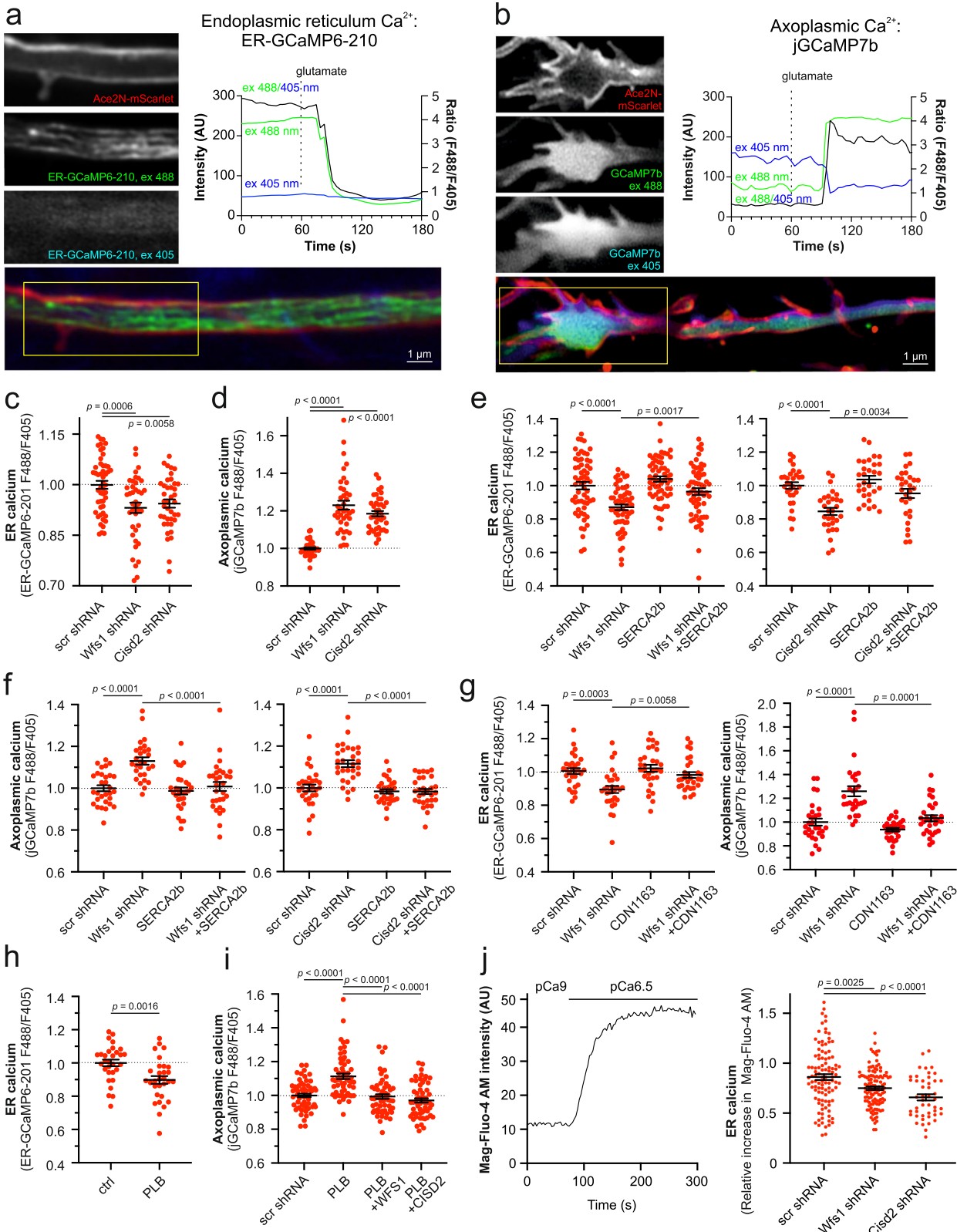

down of IP$_3$R1 and IP$_3$R3 did not normalize Ca$^{2+}$ ER/axoplasmic homeostasis in WFS1-deficient neurons. Altogether, the suppression of IP$_3$R significantly improves neither ER nor axoplasmic Ca$^{2+}$ levels in WFS1-deficient neurons.

Thus, an increased ER Ca$^{2+}$ leak through RyR channels, but not through the IP$_3$R, seems to contribute to the ER-axoplasm Ca$^{2+}$ disturbances in WFS1-deficient neurons.

## Diminished ER to mitochondria Ca$^{2+}$ flux decreases basal mitochondrial Ca$^{2+}$ levels

ER Ca$^{2+}$ depletion has been shown to impair the direct Ca$^{2+}$ flux from the ER to mitochondria at the ER-mitochondria contact sites mediated via IP$_3$R[22–24]. Indeed, WFS1- or CISD2-deficient neurons released significantly less Ca$^{2+}$ to the axoplasm when stimulated by DHPG, a metabotropic glutamate receptor agonist that induces IP$_3$ production

**Fig. 1 | Decreased ER but increased axoplasmic Ca²⁺ levels in the axons of WFS1- and CISD2-deficient neurons. a** Imaging of ER Ca²⁺ in axons. Neurons were transfected at DIV 2–3 with ER-GCamp6-210 and Ace2N-mScarlet to visualize the axonal morphology and imaged ratiometrically 6–7 days later. Note the decrease in resting ER Ca²⁺ levels after treatment with 100 μM glutamate and 10 μM glycine. **b** Axoplasmic Ca²⁺ imaging at axonal endings. Neurons were transfected at DIV 2–3 with jGCaMP7b and Ace2N-mScarlet, and imaged ratiometrically 6–7 days later. **c** Resting ER Ca²⁺ levels in axons were lower in WFS1- or CISD2-deficient neurons. $n = 40$, 50 or 50 neurons in scrambled-, *Wfs1*- or *Cisd*-shRNA expressing groups, respectively. One-way ANOVA and Šídák's multiple comparisons test. **d** Resting levels of axoplasmic Ca²⁺ measured at axonal endings were higher in WFS1- or CISD2-deficient neurons. $n = 38$, 40 or 40 neurons, Brown-Forsythe ANOVA and Dunnett's T3 multiple comparisons test. **e** Overexpression of SERCA2b restored the ER Ca²⁺ levels in WFS1-deficient (left panel) and CISD2-deficient (right panel) neurons. $n = 60$ (left), $n = 30$, 30, 30 or 29 (right) neurons, One-way ANOVA and Šídák's multiple comparisons test. **f** Overexpression SERCA2b normalized the axoplasmic

Ca²⁺ levels in WFS1-deficient (left panel) and CISD2-deficient (right panel) neurons. $n = 30$, 28, 30 or 30 (left), $n = 30$, 29, 30 or 30 (right) neurons, One-way ANOVA and Šídák's multiple comparisons test. **g** SERCA activator CDN1163 (0.5 μM for 48 h) restored ER (left panel) and axoplasmic (right panel) Ca²⁺ levels in Wfs1-deficient neurons. $n = 30$ (left), $n = 29$, 28, 29 or 30 (right) neurons, One-way ANOVA and Šídák's multiple comparisons test (left) or Brown-Forsythe ANOVA and Dunnett's T3 multiple comparisons test (right). **h** Overexpression of phospholamban (PLB) suppresses ER Ca²⁺ levels in wt neurons. $n = 30$ neurons, two-tailed *t* test. **i** The elevated axoplasmic Ca²⁺ in phospholamban-expressing neurons was normalized by overexpressing WFS1 or CISD2. $n = 60$, 60, 59, or 59 neurons, One-way ANOVA and Šídák's multiple comparisons test. **j** ER Ca²⁺ uptake is lower in outer-membrane permeabilised WFS1- and CISD2-deficient neurons. The left panel represents a sample curve. Data presented in the right panel are presented as relative Ca²⁺ uptake of the transfected cell to Ca²⁺ uptake in nontransfected neurons in the same field. $n = 104$, 119 or 46 neurons, Brown-Forsythe ANOVA and Dunnett's T3 multiple comparisons test. Data are presented as mean ± SEM.

in glutamatergic neurons (Fig. 3a). Further ratiometric measurements of mitochondrial Ca²⁺ using the mitochondrially targeted G-Cepia3mt (Fig. 3b) demonstrated that these changes were associated with lower mitochondrial Ca²⁺ uptake (Fig. 3c) and decreased resting Ca²⁺ levels in the mitochondrial matrix (Fig. 3d, see also Supplementary Fig. 4 where higher affinity sensor G-Cepia2mt was used). Note that similarly to GCamp6 and 7, GCepia probes also have two excitation peaks allowing ratiometric measurements independent of probe expression levels[25].

We also tested the efficacy of wild-type WFS1 as well as four different disease-linked mutations to rescue mitochondrial Ca²⁺ homeostasis in WFS1 knock-down neurons. The mutations A684V and P724L are associated with a moderately severe disease phenotype, whereas the mutations K836N and E864K, as outlined in an in-depth review[26], result in less severe manifestations of the disease. Our findings demonstrate that wild-type, K836N, and E864K WFS1 were able to fully rescue the effects of WFS1 knock-down on mitochondrial calcium levels (Supplementary Fig. 5). However, the A684V and P724L mutations had no rescue capacity. These results suggest that mutations associated with varying degrees of disease severity affect mitochondrial calcium dynamics to different extents. Furthermore, these experiments provide additional validation for the specificity of our knock-down model.

To assess whether the lowered matrix Ca²⁺ level depends on impaired ER function, we asked if the mitochondrial Ca²⁺ levels could be restored by enhancing SERCA activity. Indeed, overexpression of SERCA2b rescued the matrix Ca²⁺ level in WFS1-deficient axons (Fig. 3e). Similarly, inhibition of the ER Ca²⁺ leak by downregulation of RyR normalized mitochondrial Ca²⁺ (Fig. 3f). Considering that IP₃R channels participate in Ca²⁺ transfer from ER to the mitochondrial matrix[27], we overexpressed IP₃R1 (an isoform highly expressed in neuronal cells) in WFS1-deficient neurons. Only the wt and active channel fragment of IP₃R1[11] but not the pore-dead D2550A mutant of IP₃R[24] restored resting mitochondrial Ca²⁺ levels (Fig. 3g). This indicates that Ca²⁺ flux through the IP₃R is needed to restore mitochondrial Ca²⁺ dynamics and also excludes other Ca²⁺-independent properties of IP₃R, such as protein scaffolding and ER-mt contacts[24]. Interestingly, IP₃R1 overexpression also restored axoplasmic Ca²⁺ levels, demonstrating the role of mitochondrial Ca²⁺ uptake in axonal Ca²⁺ regulation (Supplementary Fig. 6). Thus, the loss of WFS1 is associated with lower IP₃R-mediated ER to mitochondria Ca²⁺ transfer leading to decreased mitochondrial Ca²⁺ content.

## Ca²⁺ transfer through MAMs is involved in the decreased mitochondrial Ca²⁺ uptake

Previous findings, relying on proximity ligation assay and electron microscopy, show a lower number of ER-mitochondrial contacts in fixed WS1 patient fibroblasts[12]. Additionally, less sensitive ER- and mitochondrial marker colocalization studies suggest less overlap

between these organelles in hiPSC-derived neurons[13]. We addressed this question further by utilizing a new-generation NanoBit-based reversible probe called MAM-tracker[28], allowing us to measure MAM status in living neurons (Fig. 4a). We also normalized the luminescence readings from NanoLuc to cytosolic iRFP720 fluorescence signal to take into account the potential toxicity and/or variation in cell count. Our results, presented in Fig. 4b, demonstrate a significant reduction in the quantity of these contact sites in WFS1- as well as in CISD2-deficient neurons.

We next asked whether the restoration of MAMs, or ER-Mito Ca²⁺ flux through the MAMs, could also improve mitochondrial Ca²⁺ levels in WFS1-deficient neurons. Indeed, the overexpression of GRP75, known to link the mitochondrial VDAC and ER-located IP₃R[22], restored the quantity of MAMs in the WFS1-deficient neurons (Fig. 4c). Moreover, it also restored mitochondrial Ca²⁺ uptake in WFS1-deficient neurons (Fig. 4d). These experiments specifically validate that the diminished Ca²⁺ transfer through MAMs is involved in the decreased mitochondrial Ca²⁺ uptake in these neurons.

## Low mitochondrial Ca²⁺ contributes to poor ATP production

It is well known that mitochondrial Ca²⁺ bursts are required to activate pyruvate dehydrogenase phosphatase, which activates pyruvate dehydrogenase (PDH), fueling the tricarboxylic acid cycle (TCA cycle) that in turn generates NADH for mitochondrial ATP production[29–32]. Therefore, we hypothesized that a lack of sufficient PDH activity in WFS1- and CISD2-deficient neurons might diminish mitochondrial ATP production and, consequently, push the neuronal metabolism toward glycolysis.

Indeed, several lines of indirect evidence suggest that a similar mechanism might be involved in WS[11,13,33]. First, the mitochondrial membrane potential, which reflects the efficiency of the NADH-powered mitochondrial respiratory chain, was slightly lower in WFS1- and CISD2-deficient neurons (Fig. 5a). Second, the axonal ATP level, which is predominantly of mitochondrial origin and depends on PDH activity (Fig. 5b middle panel) was lower in WFS1- and CISD2-deficient neurons (Fig. 5b right panel). Third, when mitochondrial ATP supply is suppressed, it promotes glycolytic ATP production, and this can be measured indirectly by assessing the axoplasmic NADH/NAD⁺ ratio[34]. Figure 5c, d demonstrate increased axoplasmic NADH/NAD⁺ ratios in the axons of both WFS1- and CISD2-deficient neurons, measured using Peredox[35] or SoNar[36]. Due to the stronger basal fluorescence signal of Peredox-mCitrine in neurons, it was preferred in further experiments.

We also wanted to confirm whether inhibition of PDH activity in neurons would lead to similar changes in axoplasmic ATP and NADH levels. Figure 6a, b demonstrate that overexpression of the PDH kinase 1, which inhibits PDH, suppressed mitochondrial ATP production, thereby increasing the axoplasmic NADH/NAD⁺ ratio, reflecting activation of glycolysis. More importantly, overexpression of Ca²⁺-

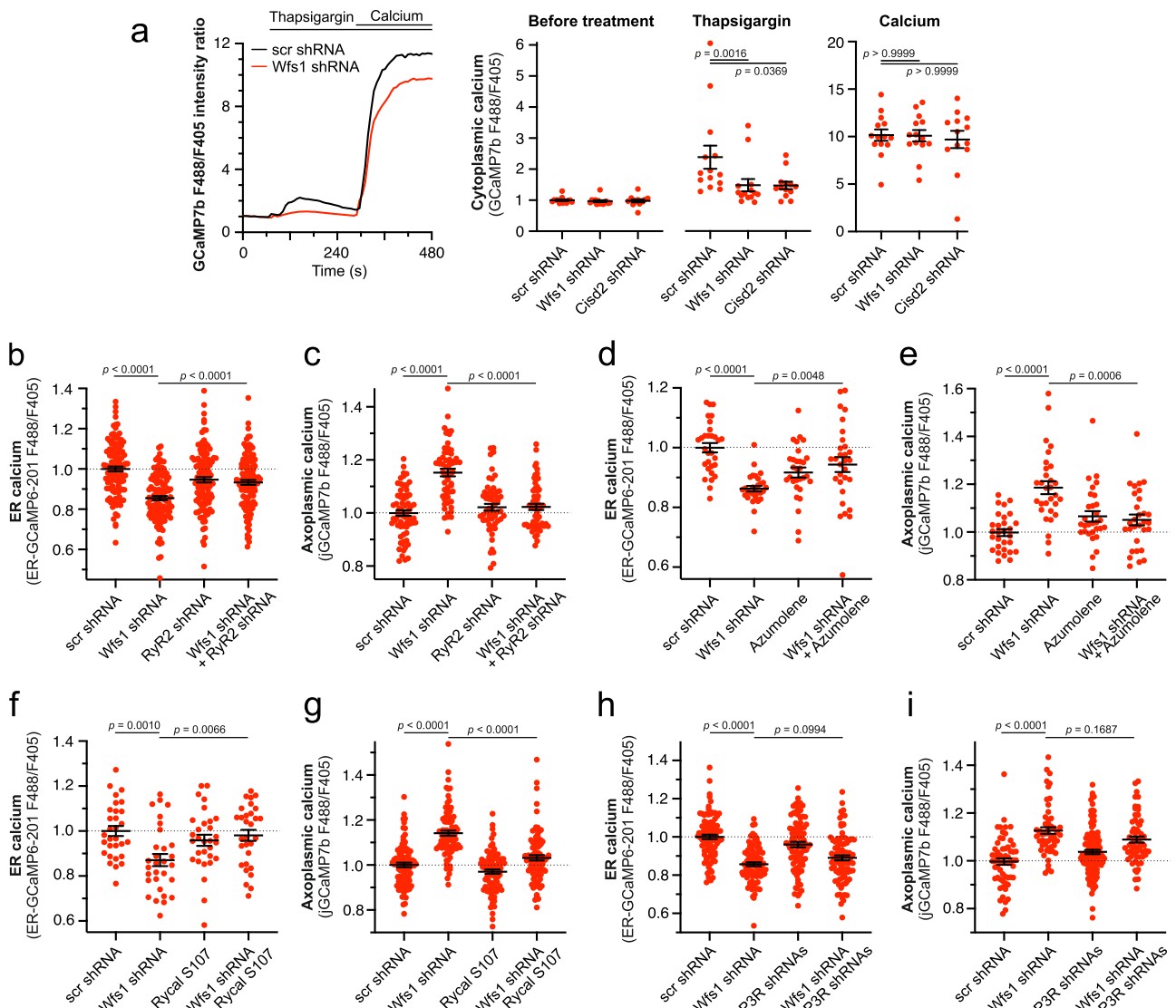

**Fig. 2 | Leak through the RyR receptor is responsible for ER Ca²⁺ loss in WFS1-deficient neurons. a** Thapsigargin-induced Ca²⁺ release from the ER to cytoplasm is lower in WFS1- and CISD2-deficient neurons. The left panel depicts the typical Ca²⁺ transients obtained after treatment with 2 μM Thapsigargin in Ca²⁺-free media in the presence of 0.5 mM EGTA and the effect of subsequent reintroduction of external 10 mM Ca²⁺. The right panel shows the peak values of these transients before and after Thapsigargin or Ca²⁺ treatments. n = 14 (scrambled and *Wfs1* shRNA) or 13 (*Cisd2* shRNA) neurons, Brown-Forsythe ANOVA and Dunnett's T3 multiple comparisons test. **b, c** RyR2 knock-down restored the ER and axoplasmic Ca²⁺ levels in the axons of the WFS1-deficient neurons. n = 120 (**b**), n = 70, 58, 58 or 59 (**c**) neurons, one-way ANOVA, and Šídák's multiple comparisons test.

**d, e** Treatment with 20 μM Azumolene (48 h) restored the basal ER and axoplasmic Ca²⁺ levels in the axons of WFS1-deficient neurons. n = 30, 29, 30 or 30 (**d**), n = 29, 30, 30 or 30 (**e**) neurons, One-way ANOVA and Šídák's multiple comparisons test. **f, g** Treatment with 5 μM Rycal S107 (48 h) restored the basal ER and axoplasmic Ca²⁺ levels in WFS1-deficient neurons. n = 30 (**f**), n = 88, 87, 90 or 89 (**g**) neurons, one-way ANOVA and Šídák's multiple comparisons test. **h, i** IP₃R1 and IP₃R3 knock-down did not restore the basal ER and axoplasmic Ca²⁺ levels in WFS1-deficient neurons. n = 90 (**h**), n = 60, 57, 118 or 60 (**i**) neurons, Brown-Forsythe ANOVA and Dunnett's T3 multiple comparisons test or Kruskal–Wallis test & Dunn's multiple comparisons test. Data are presented as mean ± SEM.

insensitive PDH phosphatase 2 (PDP2), which activates PDH, improved mitochondrial ATP production (Fig. 6c) and axoplasmic NADH/NAD⁺ levels (Fig. 6d). Thus, the bioenergetic capacity of mitochondria is compromised in WFS1-deficient neurons. However, can this be corrected by normalizing ER Ca²⁺ handling and increasing mitochondrial Ca²⁺ uptake from ER?

To test this hypothesis, we used several approaches to increase Ca²⁺ in the mitochondrial matrix and then followed ATP content in neurons. Overexpression of IP₃R1 in WFS1-deficient neurons, which restored mitochondrial Ca²⁺ content (Fig. 3g), also improved axoplasmic ATP levels (Fig. 6e). Moreover, overexpressed MIRO1, known to increase the mitochondrial Ca²⁺ uptake[37], normalized resting mitochondrial and axoplasmic Ca²⁺ levels (Fig. 6f, g) and increased

axoplasmic ATP content (Fig. 6h). Similarly, CGP37157, a well-known inhibitor of the mitochondrial Na⁺/Ca²⁺ exchanger (mNCX) that is known to prevent Ca²⁺ extrusion from the mitochondrial matrix[38], restored both the axoplasmic and mitochondrial Ca²⁺ levels (Fig. 6i, j) as well as axoplasmic ATP content (Fig. 6k). These results suggest that insufficient mitochondrial Ca²⁺ uptake could be, at least partially, a cause for the reduced ATP production in WFS1-deficient neurons.

## WFS1 and CISD2 interact functionally and compensate for each other

The alterations in Ca²⁺ homeostasis and energy metabolism described above were quite similar for both WFS1- and CISD2-deficiency, supporting a converging role for WFS1 and CISD2[39]. Nevertheless, no

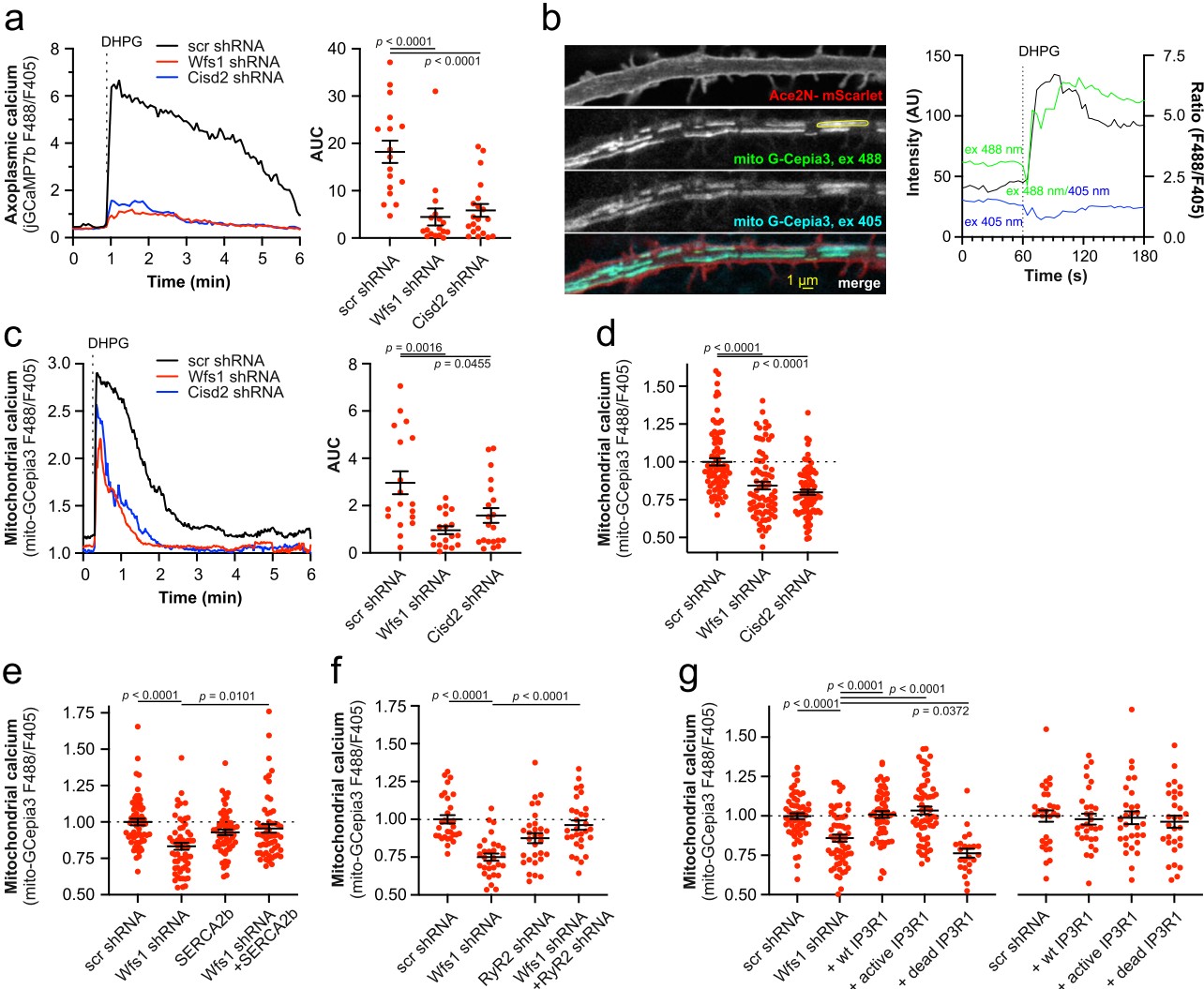

**Fig. 3 | Impaired ER Ca²⁺ homeostasis is associated with suppressed mitochondrial Ca²⁺ uptake and decreased basal mitochondrial Ca²⁺ levels. a** DHPG-induced Ca²⁺ release from ER to axoplasm is lower in WFS1- and CISD2-deficient neurons. The left panel depicts averaged Ca²⁺ transients obtained after treatment with 100 µM DHPG and the right panel shows the area under the curve (AUC) of these transients. $n = 17$, 17 or 20 neurons, One-way ANOVA & Holm-Šídák's multiple comparisons test. **b** Imaging of mitochondrial Ca²⁺ in axons. Neurons were transfected at DIV 2–3 with mitochondrially targeted GCepia3 and Ace2N-mScarlet to visualize the axonal shaft and imaged ratiometrically 5–6 days later. Note the increase in the resting mitochondrial Ca²⁺ levels after treatment with 100 µM DHPG. **c** DHPG-induced Ca²⁺ uptake to axonal mitochondria is lower in WFS1- and CISD2-deficient neurons. The left panel depicts averaged Ca²⁺ transients obtained after treatment with 100 µM DHPG and the right panel the area under the curve of these

transients. $n = 18$, 18 or 20 neurons, Brown-Forsythe ANOVA and Dunnett's T3 multiple comparisons test. **d** Basal mitochondrial Ca²⁺ levels are lower in the axons of WFS1- and CISD2-deficient neurons. $n = 80$, 76 or 80 neurons, Kruskal–Wallis test & Dunn's multiple comparisons test. **e** Overexpression of SERCA2b restores mitochondrial Ca²⁺ levels in the axons of WFS1-deficient neurons. $n = 60$, 59, 60 or 57 neurons, Kruskal–Wallis test & Dunn's multiple comparisons test. **f** Knock-down of RyR2 restores mitochondrial Ca²⁺ levels in the axons of WFS1-deficient neurons. $n = 30$ neurons, One-way ANOVA and Šídák's multiple comparisons test. **g** Overexpression of wt IP₃R1 but not pore-dead mutant D2550A restores mitochondrial Ca²⁺ levels in the axons of WFS1-deficient neurons. $n = 58$, 60, 57, 60 or 23 (left panel), $n = 30$ (right panel) neurons, Brown-Forsythe ANOVA and Dunnett's T3 multiple comparisons test. Data are presented as mean ± SEM.

physical or/and functional interaction between these proteins has been demonstrated to date. WFS1 and CISD2 are both expressed homogeneously throughout the ER network and strongly expressed in the ER in axonal termini (Fig. 7a). WFS1-RFP and CISD2-YPet colocalize perfectly in ER (Fig. 7b), proximity ligation assay shows their proximity (Supplementary Fig. 7) and computational docking model for human CISD2 and WFS1 suggests direct interaction of these proteins (Supplementary Fig. 8).

Overexpressed CISD2-myc co-immunoprecipitated with EGFP-WFS1 in HEK293 cells (Fig. 7c). Also, the endogenous WFS1 strongly co-immunoprecipitates with pulled-down CISD2 in HEK cells (Fig. 7d). Notably, both endogenous WFS1 and CISD2 co-immunoprecipitate with RyR2 (Fig. 7e) as well as with SERCA2

showing that both are present in complexes with SERCA2 and RyR2 (Fig. 7f).

Next, we assessed whether CISD2 could compensate for the loss of WFS1 and vice versa, thereby determining whether WFS1 and CISD2 could functionally complement each other. Indeed, CISD2 overexpression restored the axoplasmic Ca²⁺ homeostasis and ATP levels in neurons lacking WFS1 (Fig. 8a, b left panels), while WFS1 overexpression did the same in neurons lacking CISD2 (Fig. 8a, b right panel).

In addition, mitophagy, mitochondrial morphology and mitochondrial density, which were all impaired in WFS1- or CISD2-deficient neurons, were partially restored upon respective overexpression of CISD2 or WFS1 in these neurons (Fig. 8c, d, Supplementary Fig. 9).

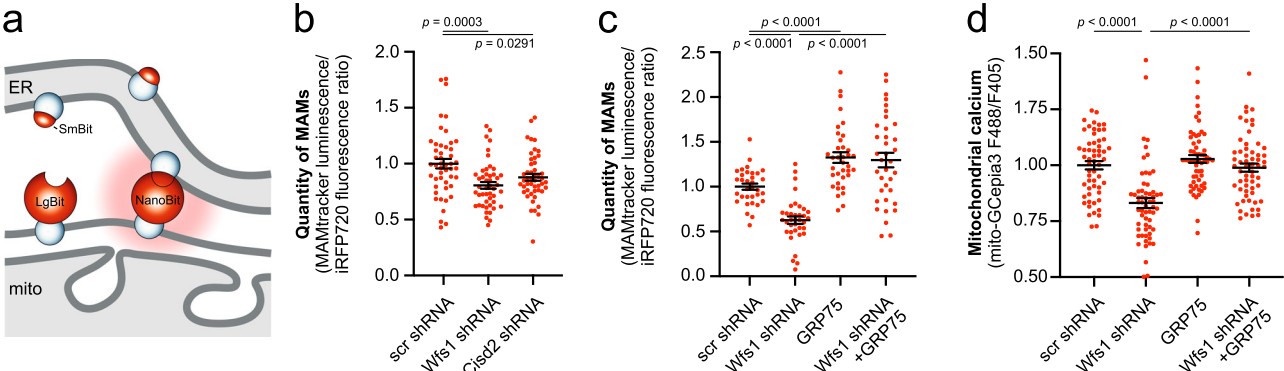

**Fig. 4 | Reduced quantity of MAMs in WFS1- and CISD2-deficient neurons.**
**a, b** Quantity of MAMs is lower in WFS1- or CISD2-deficient neurons. Neurons were transfected at DIV 2–3 with MAMTracker-LUC, iRFP720, and scrambled, *Wfs1*- or *Cisd2* shRNA. On the fifth day post-transfection, culture media were replaced with Nano-Glo® Live Cell Reagent, and luminescence and fluorescence were recorded. Luminescence for the complemented NanoBiT was normalized to cytosolic iRFP720 fluorescence to exclude differences in cell count. $n = 48, 48$ or 47 wells of

neurons, Ordinary one-way ANOVA followed by Dunnett's multiple comparisons test. **c** Overexpression of GRP75 restored the MAMs in WFS1- or CISD2-deficient neurons. $****P < 0.0001$, $n = 33, 36, 36$ or 36 wells of neurons, Brown-Forsythe ANOVA test followed by Dunnett's T3 multiple comparisons test. **d** Overexpression of GRP75 restored the mitochondrial $Ca^{2+}$ levels in the axons of WFS1-deficient neurons. $****P < 0.001$, $n = 57, 60, 60$ or 59 wells of neurons, Kruskal–Wallis test & Dunn's multiple comparisons test. Data are presented as mean ± SEM.

It is known that both bioenergetic and reductive stress affect axonal growth and regeneration[11,40]. Improved cellular energetics promote axonal growth, and sufficient $NAD^+$ levels are essential to keep axons alive. Figure 8e demonstrates that axons grew more slowly in WFS1- and CISD2-deficient cortical neurons. This deficit was restored by overexpressing SERCA2b, indicating that a defect in axonal development is directly related to initial $Ca^{2+}$ handling deficits at the level of the ER (Supplementary Fig. 10). Notably, axonal length was partially restored upon overexpression of CISD2 or WFS1 in these settings (Fig. 8e).

Thus, these data demonstrate that WFS1 and CISD2 are involved in the same pathway, either directly or by interacting with common interaction partners.

## Pharmacological intervention improves mitochondrial and axonal health

Altogether, our results reveal that alterations in mitochondrial $Ca^{2+}$ homeostasis, which depends on ER function, appear to be a key element in the pathological mechanisms of WFS1- and CISD2-deficiency. Theoretically, this homeostasis could be restored by pharmacological interventions aiming to increase ER $Ca^{2+}$ uptake and release and/or enhance $Ca^{2+}$ uptake by mitochondria. We therefore next tested whether we could improve mitochondrial health in WFS1-deficient neurons using drugs known to prevent $Ca^{2+}$ depletion in ER and mitochondria. These drugs were the SERCA activator CDN1163 (0.5 μM), an inhibitor of $Ca^{2+}$ release via RyR azumolene (20 μM), an inhibitor of $Ca^{2+}$ leak through the RyR Rycal S107 (5 μM) and a selective inhibitor of the mitochondrial $Na^+/Ca^{2+}$ exchanger CGP37157 (10–20 μM). We also included liraglutide (500 nM) in our experiments as it is a first-line treatment in WS patients and it is neuroprotective in animal experiments[7,41,42]. We examined the effects of these drugs on selected parameters of mitochondrial and axon health (mitochondrial density and length, mitophagy intensity, ATP/ADP ratio and axonal length). The results summarized in Table 1 and Supplementary Fig. 11 were very encouraging since they demonstrate that all of these treatments were protective in most settings, with CGP37157 being effective in all settings. These data clearly suggest that ER/mitochondrial $Ca^{2+}$ homeostasis plays an important role in the pathogenesis of mitochondrial disturbances in WFS1-deficiency and correction of this homeostasis could be an effective tool to normalize neuron health in WS.

## Discussion

This study unveils three pivotal findings. Firstly, we show that specifically the loss of ER calcium is responsible for reduced $IP_3R$-mediated

$Ca^{2+}$ transfer from the ER to the mitochondria in WS neurons. This loss is induced by diminished SERCA activity and concurrent $Ca^{2+}$ leakage via RyR receptors. Secondly, our investigation reveals that diminished $Ca^{2+}$ influx into the mitochondria is hindering $Ca^{2+}$-dependent pyruvate dehydrogenase and Krebs cycle turnover, compromising mitochondrial ATP production and directing the neurons toward glycolysis. Thirdly, our study also demonstrates that WFS1 and CISD2, representing sensor systems for ER- and redox stress, are integral components of the same protein complex, regulating SERCA and RyR in a complementary manner. Collectively, these findings provide valuable insights into the underlying mechanisms of WS and offer new potential avenues for therapeutic intervention.

The importance of WFS1 in ER $Ca^{2+}$ signaling has become increasingly evident through various lines of research. Reductions in releasable ER $Ca^{2+}$ content have been observed in WFS1 knockout HEK293 cells[8], pancreatic β cells lacking WFS1[9], rat insulinoma cells[43] and fibroblasts from individuals with WS[12]. The ER $Ca^{2+}$ content has also been shown to decrease in MEFs and cardiomyocytes of CISD2 KO mice[16,44], although no change or even an increase in ER $Ca^{2+}$ levels has also been reported[45–49]. The decrease in ER $Ca^{2+}$ content has been linked to reduced SERCA activity[44,50,51] and $Ca^{2+}$ leakage through the RyR[2,10], offering a rationale for considering the repurposing of the established drug dantrolene as a potential treatment for WS by inhibiting RyR[10,52]. However, recent reports[11,12,14,53], suggest that the primary issue may be suppressed $IP_3R$-mediated $Ca^{2+}$ efflux, offering therapeutic options involving sigma-1 receptor agonists known to activate this process[14]. Nevertheless, this explanation doesn't clarify why ER $Ca^{2+}$ content decreases, as it should increase in that case. It also does not explain the increase in basal axoplasmic $Ca^{2+}$ concentrations observed in many earlier reports[10–12].

In our study, we adopted a systematic approach to assess $Ca^{2+}$ levels within the ER, cytosol, and mitochondria, along with monitoring ER-to-mitochondria $Ca^{2+}$ flux in primary neurons. Our objective was to delineate alterations in $Ca^{2+}$ homeostasis in WS and identify potential targets for correction. Our results confirm that the primary deficit in neurons lacking WFS1 or CISD2 is significantly reduced ER $Ca^{2+}$ store content in axons. We demonstrate that this reduction can be attributed to a combination of decreased SERCA activity and an increase in $Ca^{2+}$ leakage through RyR channels. ER $Ca^{2+}$ levels were restored by the overexpression of SERCA or its activation with a small-molecule allosteric activator, CDN1163. Similarly, knock-down or inhibition of RyR channels by a dantrolene analog normalized the ER $Ca^{2+}$ store content. Importantly, a relatively specific RyR leak inhibitor, Rycal

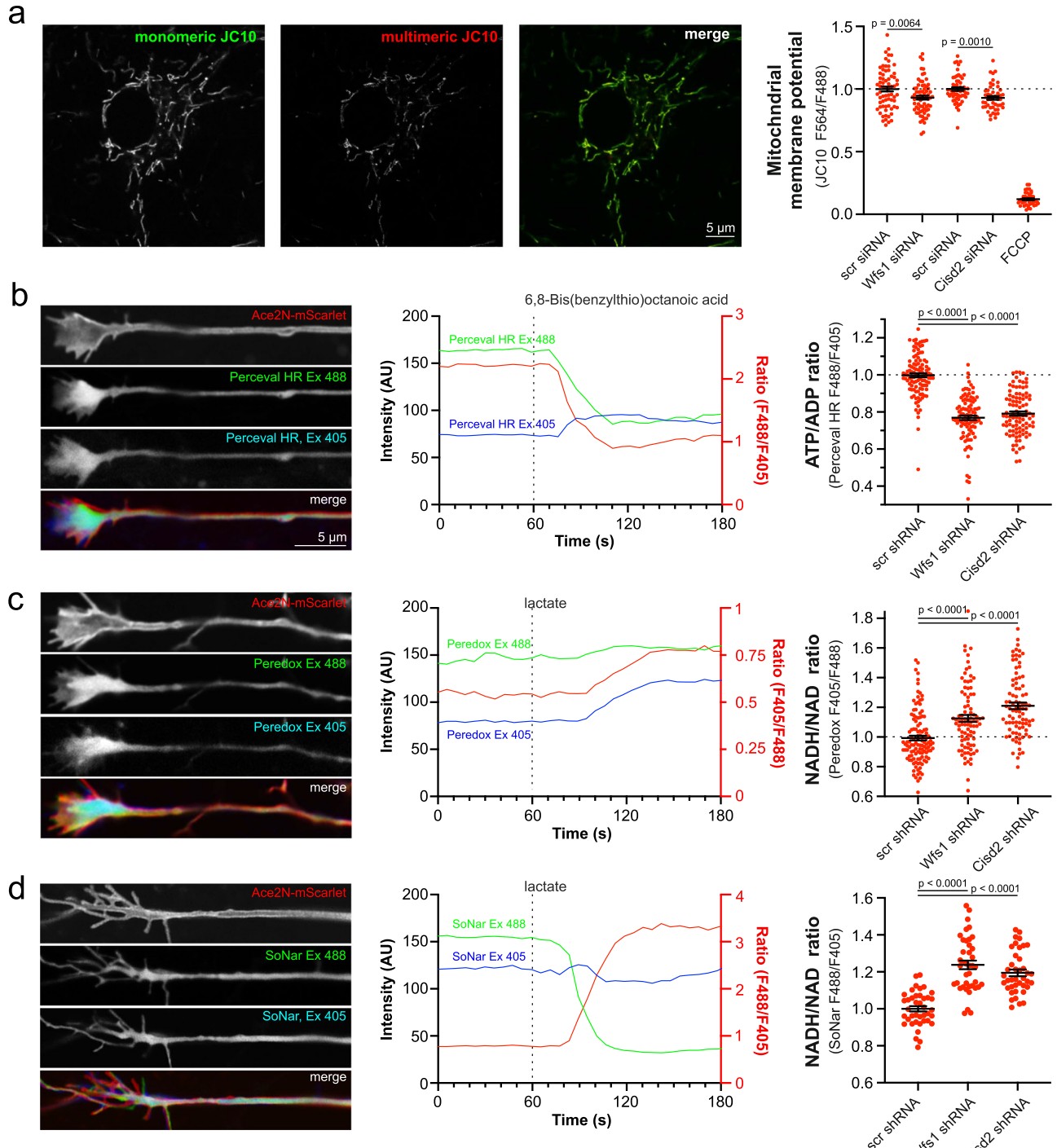

**Fig. 5 | Decreased mitochondrial membrane potential, ATP production and increased NADH/NAD⁺ ratio in the axons of WFS1- and CISD2-deficient neurons.** **a** Mitochondrial membrane potential is lower in WFS1- and CISD2-deficient neurons. Neurons were transfected at DIV 2 siRNA against *Wfs1* and *Cisd2* and stained and visualized 48–72 h later. $n = 70, 70, 56, 48$ or $40$ fields of neurons, two-tailed unpaired t-test. Note that *Wfs1-* and *Cisd2* siRNAs were tested in separate experiments. **b** Axoplasmic ATP levels are lower in WFS1- and CISD2-deficient neurons. Neurons were transfected at DIV 2–3 with Perceval HR and the ATP/ADP ratio was visualized at axonal endings at DIV 8–9. Note the decrease in ATP/ADP ratio after

inhibition of mitochondrial pyruvate dehydrogenase by 50 µM 6,8-Bis(benzylthio) octanoic acid. $n = 109, 100$ or $98$ neurons, Kruskal–Wallis test & Dunn's multiple comparisons test. **c, d** NADH/NAD⁺ ratio is increased in the axoplasm of WFS1- and CISD2-deficient neurons. Neurons were transfected at DIV 2–3 with Peredox (C) or SoNar (D) and the NADH/NAD⁺ ratio was visualized at axonal endings at DIV 8–9. Note the decrease in NADH/NAD⁺ ratio after treatment with 10 mM lactate. $n = 119, 89$ or $88$ neurons, Kruskal–Wallis test & Dunn's multiple comparisons test (**c**) or $n = 40$ neurons, Brown-Forsythe ANOVA and Dunnett's T3 multiple comparisons test (**d**). Data are presented as mean ± SEM.

S107, produced similar corrective effects on ER Ca²⁺ store. This reduced ER Ca²⁺ uptake and increased Ca²⁺ leak from the ER also had a direct impact on axoplasmic Ca²⁺ levels, resulting in the accumulation of Ca²⁺ within the axoplasm.

A diminished availability of ER Ca²⁺ is a primary factor contributing to the reduced ER-to-mitochondria Ca²⁺ flux. Our findings demonstrate that the absence of WFS1 or CISD2 is linked to a decreased response in agonist-evoked, IP₃R-mediated Ca²⁺ transfer

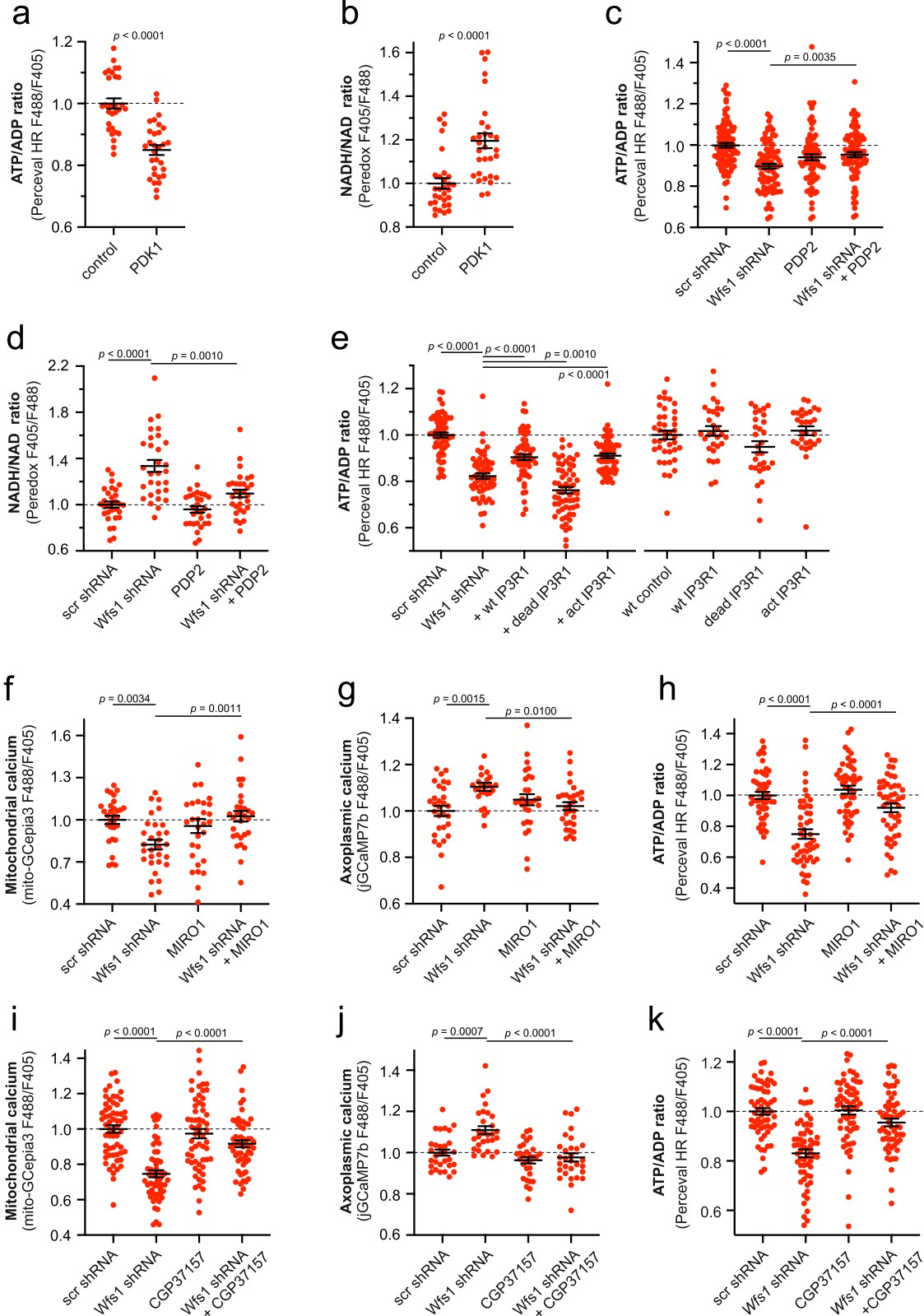

from the ER to the mitochondria. Consequently, this results in a decline in mitochondrial $Ca^{2+}$ uptake, impacting both the $Ca^{2+}$ peaks and basal mitochondrial $Ca^{2+}$ levels. Most notably, our study establishes that the depletion of ER $Ca^{2+}$ stands as the primary contributor to mitochondrial $Ca^{2+}$ loss. The restoration of mitochondrial $Ca^{2+}$ transfer and levels was achieved through the overexpression/activation of SERCA2b and the suppression/inhibition of RyR2. Remarkably, the ER-to-mitochondria interaction predominantly occurs via the IP$_3$R, as the overexpression of the latter successfully reinstated mitochondrial $Ca^{2+}$ levels. This elucidates the seemingly contradictory roles of RyRs and IP$_3$Rs in WS; while RyRs appear to be responsible for ER $Ca^{2+}$ leaks and exhibit a detrimental effect, the role of IP$_3$Rs is to facilitate the transfer of ER $Ca^{2+}$ to the mitochondria, proving to be beneficial in this context.

**Fig. 6 | Low mitochondrial Ca²⁺ is related to poor ATP production.**
**a**, **b** Suppression of pyruvate dehydrogenase by overexpressing the pyruvate dehydrogenase kinase 1 decreases axoplasmic ATP levels and increases axoplasmic NADH levels. $n = 30$ neurons, two-tailed unpaired $t$ test (**a**) or two-tailed Mann–Whitney test (**b**). **c** Overexpression of PDP2 partially restores the ATP levels at the axonal endings of WFS1-deficient neurons. $n = 90, 90, 80$ or 89 neurons, One-way ANOVA and Šídák's multiple comparisons test. **d** Overexpression of PDP2 normalizes the NADH/NAD⁺ ratio at the axonal endings of WFS1-deficient neurons. $n = 30$ neurons, Brown-Forsythe ANOVA and Dunnett's T3 multiple comparisons test. **e** IP₃R1 overexpression restored the ATP levels in the axoplasm of WFS1-deficient neurons. $n = 60, 60, 60, 60$ or 59 (left panel), $n = 40, 30, 30, 30, 30$ (right

panel) neurons, one-way ANOVA and Šídák's multiple comparisons test. **f–h** Overexpressed MIRO1 normalized the mitochondrial (**f**) and axoplasmic Ca²⁺ (**g**) and increased the axonal ATP content (**h**) in WFS1-deficient neurons. $n = 30$ (**f**), $n = 30, 27, 29, 30$ (**g**), $n = 50, 50, 50, 49$ (**h**), One-way ANOVA and Šídák's multiple comparisons test. **i–k** Treatment with CGP37157 (10 μm for 48 h) normalized the mitochondrial (**i**) and axoplasmic Ca²⁺ (**j**) and increased the axoplasmic ATP content (**k**) in WFS1-deficient neurons. $n = 60, 60, 60, 58$ (**f**), $n = 30, 29, 30, 30$ (**g**), $n = 60$ (**h**), Brown-Forsythe ANOVA and Dunnett's T3 multiple comparisons test (**i**) or One-way ANOVA and Šídák's multiple comparisons test (**j**, **k**). Data are presented as mean ± SEM.

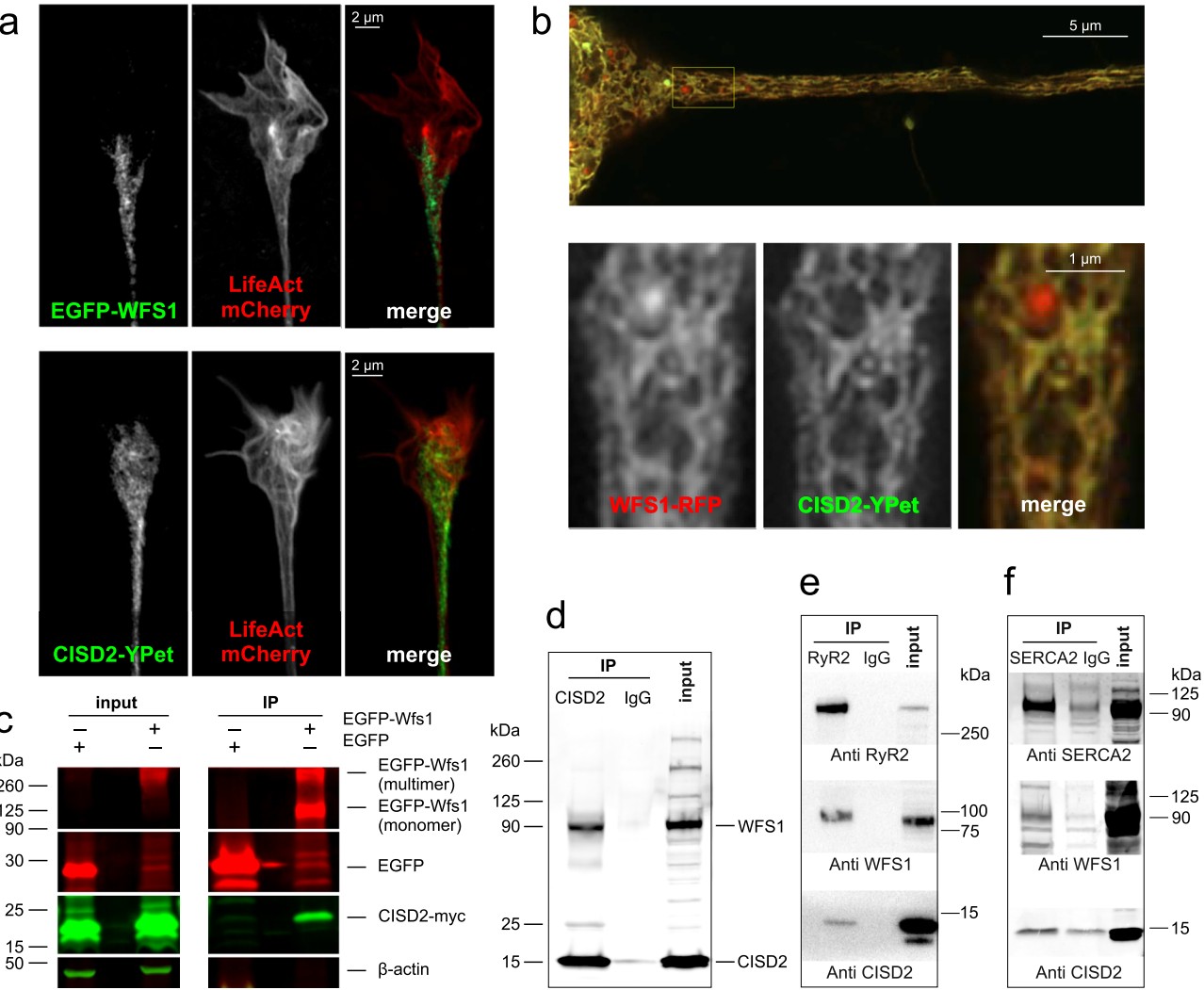

**Fig. 7 | WFS1 colocalizes and interacts with CISD2. a** Airyscan2 images demonstrating strong expression of EGFP-WFS1 and CISD2-YPet in the axonal terminals. **b** Airyscan2 image demonstrating colocalization of CISD2-YPet and WFS1-RFP in the axonal ER. Note that WFS1-RFP tends to form aggregates that do not colocalize with CISD2. **c** Overexpressed CISD2 co-immunoprecipitates with overexpressed WFS1. EGFP-WFS1 was immunoprecipitated from the HEK 293 cell lysate using GFP-Trap, and co-immunoprecipitated CISD2-myc was detected using anti-myc antibodies. **d** Endogenous WFS1 co-immunoprecipitates with endogenous CISD2. CISD2 was

immunoprecipitated from HEK293 cell lysate using mouse anti-CISD2 and detected using rabbit anti-WFS1. **e** WFS1 and CISD2 co-immunoprecipitate with RyR2. RyR2 was immunoprecipitated from lysates of 6-month-old whole mouse brains, followed by assessment of WFS1 and CISD2 presence in the immunocomplex. **f** WFS1 and CISD2 co-immunoprecipitate with SERCA2. SERCA2 was immunoprecipitated from PC6 cell lysate with mouse anti-SERCA2 antibodies and probed with rabbit anti-WFS1, anti-CISD2, and anti-SERCA antibodies.

The intriguing question also arises as to what extent the loss of MAMs observed in WFS1 and CISD2-deficient cells contributes to diminished ER-mitochondrial Ca²⁺ transfer. Overexpression of GRP75, linking the IP3R with VDAC, increases the number of MAMs and also improves the mitochondrial Ca²⁺ uptake in the WFS1-deficient neurons,

suggesting the involvement of MAM deficiency in WS. Even if MAM deficiency is not the primary cause behind diminished ER-to-mitochondria Ca²⁺ transfer, these data suggest that directly increasing MAMs can be an effective strategy to improve ER-mitochondrial Ca²⁺ transfer.

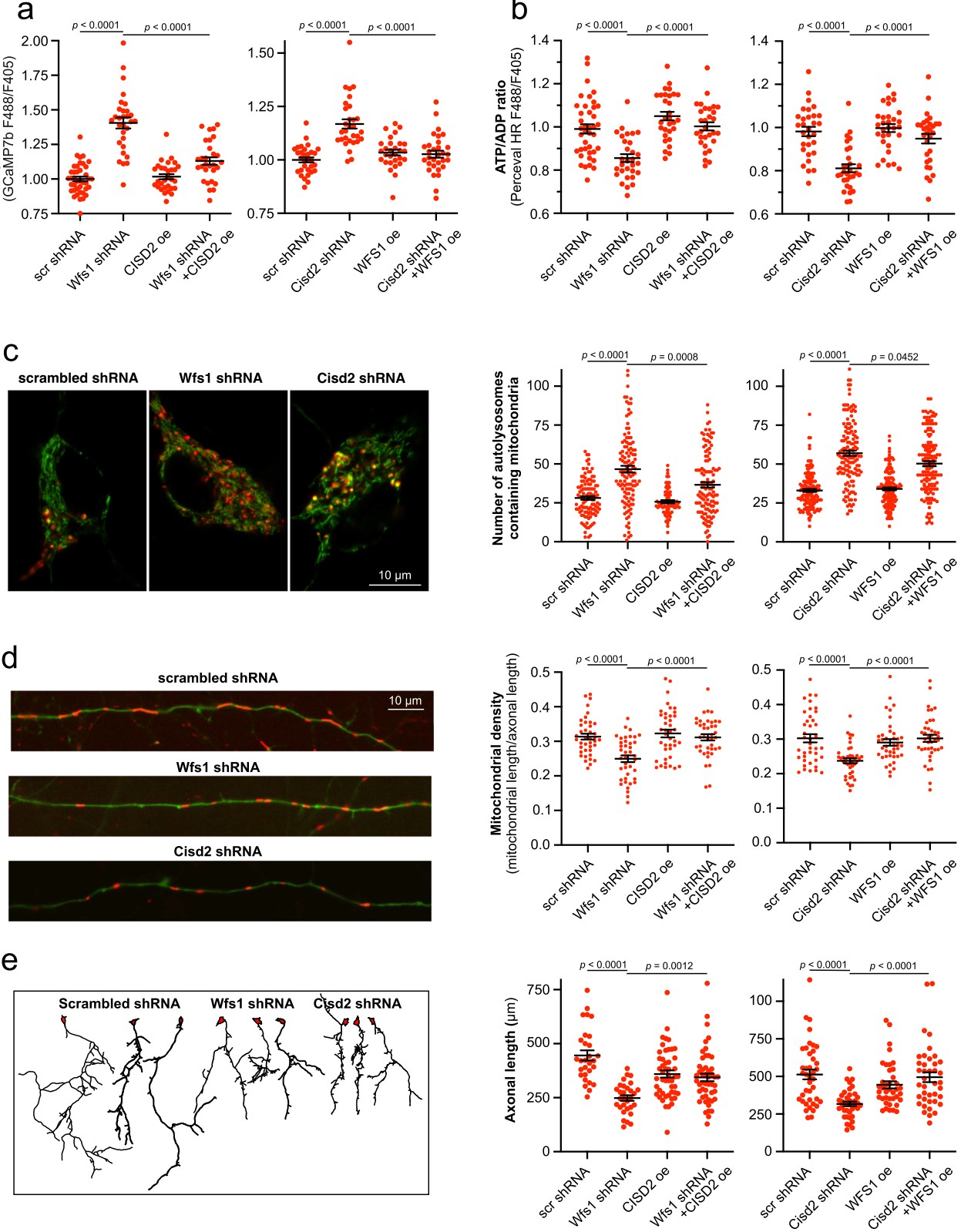

Our data shed light on how disruptions in ER Ca$^{2+}$ handling can adversely impact mitochondrial function. Within the mitochondrial matrix, several enzymes are sensitive to Ca$^{2+}$ levels, such as pyruvate dehydrogenase and Krebs cycle dehydrogenases, playing pivotal roles in generating reducing equivalents essential for the electron transport chain, thereby maintaining mitochondrial membrane potential and ATP production within the mitochondria[31,32,54]. Our experiments suggest that the sole suppression of PDH activity is sufficient to inhibit mitochondrial ATP production in axons, whereas activating PDH can restore normal mitochondrial ATP production in WFS1- and CISD2-deficient neurons. These results also indirectly suggest that a decline in Ca$^{2+}$-dependent production of reducing equivalents might be the underlying mechanism for impaired ATP production when mitochondrial matrix Ca$^{2+}$ levels are depleted.

**Fig. 8 | WFS1 and CISD2 compensate for each other. a** Overexpression of CISD2 in WFS1-deficient neurons (left panel) and WFS1 in CISD2-deficient neurons (right panel) suppress the basal axoplasmic $Ca^{2+}$ back to normal. $n = 40$, 30, 30, or 29 (left panel), $n = 31$, 30, 29, 31, (right panel) neurons, One-way ANOVA & Šídák's multiple comparisons test. **b** Overexpression of CISD2 in WFS1-deficient neurons (left panel) and WFS1 in CISD2-deficient neurons (right panel) restore normal ATP levels. $n = 40$, 30, 30, or 30 (left panel), $n = 30$, 31, 30, 30 (right panel) neurons, One-way ANOVA & Šídák's multiple comparisons test. **c** Increased mitophagy in WFS1- and CISD2-deficient neurons (left panel) is partially suppressed when overexpressing CISD2 (middle panel) or WFS1 (right panel). Primary cortical neurons were transfected with mitochondrially targeted Keima (which changes its excitation spectrum under acidic conditions: green, mitochondria at neutral pH and red, mitochondria

under acidic pH) and plasmids of interest. $n = 110$, 119, 70, or 120 (left panel), $n = 140$ neurons, Kruskal–Wallis test & Dunn's multiple comparisons test. **d** Mitochondrial loss in WFS1- and CISD2-deficient neurons (left panel) is reversed when over-expressing CISD2 (middle panel) or WFS1 (right panel). Neurons were transfected with with EGFP, mitochondrial DsRed2 and plasmids of interest. $n = 40$ neurons, One-way ANOVA & Šídák's multiple comparisons test or Brown-Forsythe ANOVA & Dunnett's T3 multiple comparisons test. **e** Suppressed axonal growth in WFS1- and CISD2-deficient neurons (left panel) is reversed when overexpressing CISD2 (middle panel) or WFS1 (right panel). Neurons were transfected with pAAV-hSyn-DsRedExpress and plasmids of interest at DIV1 and visualized at DIV3. $n = 29$, 30, 49, or 49 (left panel), $n = 40$, 38, 39, 40 (right panel) neurons, Kruskal–Wallis test & Dunn's multiple comparisons test. Data are presented as mean ± SEM.

---

Indeed, prior research has demonstrated that skeletal muscle cells lacking the mitochondrial calcium uniporter (MCU) exhibit impaired oxidative phosphorylation due to reduced mitochondrial $Ca^{2+}$ influx[55]. This indicates that the loss of mitochondrial $Ca^{2+}$ uptake significantly hampers the utilization of pyruvate by mitochondria, favoring its conversion into lactate. In alignment with these findings, our results reveal a slight mitochondrial depolarization and significantly lower ATP levels in the axons of both WFS1- and CISD2-deficient neurons. The reduction in ATP levels was restored through the activation of mitochondrial PDH, increasing mitochondrial $Ca^{2+}$ levels directly (via inhibition of the $Na^+/Ca^{2+}$ exchanger or overexpression of MIRO1), or by reestablishing the $Ca^{2+}$ flow from the ER to the mitochondria by restoring ER $Ca^{2+}$ homeostasis.

Thus, these results suggest that the loss of ER-mitochondrial $Ca^{2+}$ flux in WFS1- or CISD2-deficient neurons leads to compromised mitochondrial function. Intriguingly, a similar phenomenon has been observed in a Charcot-Marie-Tooth (CMT) disease 4 A model, where GDAP1 silencing disrupted ER-mitochondria contact sites, leading to lower mitochondrial $Ca^{2+}$ levels and the inhibition of PDH[56]. Elevated cytoplasmic $Ca^{2+}$ levels could have various consequences, potentially contributing to calpain hyperactivation observed in WS[10].

Our findings also offer insights into the functions of WFS1 and CISD2. It has been previously observed, both by us and others, that the downregulation of WFS1 or CISD2 results in similar phenotypic outcomes, despite these two proteins belonging to distinct protein families with unique characteristics[11,15,16]. WFS1, a 100-kDa tetrameric integral endoplasmic reticulum glycoprotein with nine transmembrane regions, stands in contrast to CISD2, which is a 15-kDa single-pass transmembrane ER protein housing a redox-active 2Fe-2S cluster domain within its cytosolic CDGSH domain and forming dimers[45,57–59].

Relevantly, both WFS1 and CISD2 are molecular partners of SERCA2 and RyR[44,50] (our data). Considering their notable structural differences, it is plausible to propose that these proteins mediate distinct cellular signals. WFS1 is implicated as one of the downstream components of the unfolded protein response[60,61]. CISD2, housing iron-sulfur cluster, on the other hand, appears to be involved in responding to redox stress[59,62]. Consequently, it is reasonable to speculate that WFS1 and CISD2, acting as sensors for ER stress and redox stress, respectively, synergize to integrate these converging signals, thereby regulating cellular ER calcium handling. In this scenario, we could expect that the loss of one pathway, whether it be WFS1 or CISD2, could potentially be mitigated through the overexpression of an alternative pathway. Importantly, deficiencies in $Ca^{2+}$ homeostasis and mitochondrial phenotype were corrected when CISD2 or WFS1 were overexpressed in WFS1 or CISD2-deficient neurons, respectively. Moreover, this assumption is confirmed by our experiments, where WFS1 or CISD overexpression rescued SERCA from phospholamban-induced inhibition. Thus, our study demonstrates that WFS1 and CISD2, serving as sensor systems for ER- and redox stress, regulate the SERCA and RyR in a complementary manner.

Our data identify pharmacological targets that could be adjusted to normalize various $Ca^{2+}$-dependent mitochondrial processes and

enhance neuronal health in WS (Fig. 9). We demonstrate that drugs enhancing ER $Ca^{2+}$ uptake (CDN1163) or suppressing its leakage (dantrolene analog, Rycal S107) restore normal cellular functions. This provides further rationale for investigating dantrolene, despite recent phase Ib/IIa clinical trials not yielding the expected results[52], as well as for continued clinical development of Rycal S107. Our findings also align with recent discoveries suggesting that WS could benefit from activating $Ca^{2+}$ flux from the ER through the activation of the sigma-1 receptor (S1R). In a recent study, Delprat and colleagues[14] found that the S1R agonist PRE-084 restored $IP_3R$ $Ca^{2+}$ ion transfer and mitochondrial respiration in vitro while alleviating behavioral symptoms observed in zebrafish and mouse models of WS. Our results complement this, suggesting that it is primarily the ER $Ca^{2+}$ handling capacity, rather than the integrity of mitochondria-associated membranes, that is compromised in WS. Activating $IP_3R$ may more effectively channel $Ca^{2+}$ from the ER to mitochondria, aiding cells in coping with this situation.

Our data also suggest that inhibiting $Ca^{2+}$ loss from the mitochondrial matrix via the $Na^+/Ca^{2+}$ exchanger is protective. This normalization encompasses all mitochondrial/energetic parameters affected by WS, including mitochondrial density and length, mitophagy, axonal ATP/ADP ratio, and axonal length. This strongly implies that mitochondrial $Ca^{2+}$ is the primary cause of bioenergetic deficits rather than other factors.

Notably, the GLP-1 agonist liraglutide also improved $Ca^{2+}$ homeostasis and mitochondrial health in Wfs1-deficient neurons. Liraglutide has demonstrated neuroprotective effects in animal experiments[41,42,63,64] and has become one of the first-line treatments for WS. Liraglutide, being an analog of GLP-1, binds to the Gs-coupled GLP-1 receptor present in both β-cells and neurons, stimulating adenylate cyclase to generate cAMP[65,66]. In turn, cAMP directly potentiates $IP_3$-evoked $Ca^{2+}$ release[67,68], thereby improving mitochondrial function in WFS1-deficient neurons.

Thus, the data obtained in the present study clearly show that compounds affecting ER to mitochondrial $Ca^{2+}$ flux or mitochondrial matrix $Ca^{2+}$ levels could improve mitochondrial function and neuronal health. WS serves here as an important disease prototype to identify drugs and molecules affecting ER and mitochondrial $Ca^{2+}$ fluxes. Thus, our results could open an important perspective in treating neuronal diseases associated with impaired ER-mitochondria contact sites, Parkinson's disease, Amyotrophic lateral sclerosis, Wolfram syndrome and other rare neurodegenerative or neurodevelopmental diseases.

## Methods
### Reagents and resources
Reagents and resources used in this work are given in the Supplementary Methods and Supplementary Table 1.

### Cell culture and transfection
Primary cultures of rat cortical neurons were prepared from <1-day-old neonatal Wistar rats as described earlier. Neurons in BME supplemented with 10% FBS, 2 mM GlutaMAX™-I and 100 μg/ml gentamicin

**Table 1 | Effects of different compounds on mitophagy, mitochondrial density, ATP/ADP ratio, mitochondrial length, and axonal development**

| | CDN1163 | Azumolene | Rycal S107 | CGP37157 | Liraglutide |
|---|---|---|---|---|---|
| Mitochondrial density | 99.8 ± 23.3% (p = 0.0061)[a] | 107.8 ± 14.9% (p < 0.0001)[a] | 62.2 ± 11.8% (p = 0.0025)[a] | 137.8 ± 22.8% (p < 0.0001)[a] | 136.7 ± 26.4% (p = 0.0009)[a] |
| Mitochondrial length | 59.5 ± 10.1% (p = 0.0001)[a] | 41.6 ± 11.0% (p = 0.0155)[a] | 88.8 ± 11.4% (p < 0.0001)[a] | 49.5 ± 11.9% (p = 0.0116)[a] | 66.9 ± 10.0% (p < 0.0001)[a] |
| Mitophagy | 8.9 ± 12.1% (p > 0.9999)[a] | 53.0 ± 13.6% (p < 0.0001)[a] | 17.8 ± 13.5% (p = 0.9103)[a] | 102.8 ± 5.1% (p < 0.0001)[a] | 12.1 ± 11.6% (p = 0.8878)[a] |
| ATP/ADP ratio | 80.4 ± 7.2% (p < 0.0001)[b] | 49.6 ± 9.4% (p = 0.0004)[b] | 55.6 ± 8.5% (p < 0.0001)[c] | 73.6 ± 9.1% (p < 0.0001)[b] | 49.1 ± 10.7% (p = 0.0197)[a] |
| Axonal length | 107.4 ± 21.1% (p = 0.0017)[a] | 33.3 ± 18.6% (p > 0.999)[a] | 59.0 ± 13.9% (p = 0.0023)[a] | 73.7 ± 16.5% (p = 0.0008)[a] | 96.0 ± 24.2% (p = 0.0038)[a] |

The data show the extent to which each compound rescues the Wfs1 shRNA effect towards control levels. A value of 100% represents complete rescue to the control levels, while 0% indicates no rescue from the shRNA effect. The data used for calculations are presented in detail in Supplementary Fig. 11. Data are presented as mean ± SEM.
[a]Kruskal–Wallis test followed by Dunn's multiple comparisons test.
[b]One-way ANOVA followed by Šidák's multiple comparisons test.
[c]Brown-Forsythe ANOVA folowed by Dunnett's T3 multiple comparisons test.

were plated onto 35-mm glass-bottomed dishes (MatTek, MA, USA), which were pre-coated with poly-L-lysine, at a density of $10^6$ cells per dish in 2 ml of cell suspension. After incubating for 3 h, the medium was changed to Neurobasal™ A medium containing B-27™ supplement, 2 mM GlutaMAX™-I and 100 μg/ml gentamicin. For cell transfection at DIV 2–3, the conditioned medium was replaced with 100 μl Opti-MEM I medium containing 2% Lipofectamine™ 2000 and 1–2 μg of total DNA containing an equal amount of each different plasmid. Cell dishes were incubated for 3 h, after which fresh growth medium was added. In experiments using specific shRNAs, the same amount of scrambled shRNA was transfected into the control group. In overexpression experiments, empty vector or firefly luciferase was used to compensate DNA load in control groups. Chemicals or drugs were added to transfected cells as indicated in the Figures or Figure legends. Concentrations used were chosen according to the literature.

HEK293 (ATCC CRL-1573) cells were from American Type Culture Collection and PC6-3 (Cellosaurus RRID: CVCL_7101) cell line was gift from Dan Lindholm. HEK293 cells were cultivated in DMEM medium supplemented with 10% FBS on poly-L-lysine-coated 100-mm plastic dishes. For immunoprecipitation experiments, HEK293 cells were transfected in 60-mm plastic dishes as described above, but the volume of the transfection mixture was increased by a proportionate amount of Lipofectamine™ 2000 and DNA.

### Live-cell confocal microscopy

Confocal and Airyscan imaging were performed on either a Zeiss LSM 780 or Zeiss LSM 980 with Airyscan2 Axio observer Z1 microscopes. Images were acquired with C-Apochromat ×63/1.2 water or C Plan-Apochromat ×63/1.4 oil objective and controlled by ZEN black 2.1 SP1 or ZEN blue 3.1 software. For live-cell experiments, if not stated otherwise, neurons were imaged in Neurobasal™A medium without phenol red at 37 °C and 5% $CO_2$. Time-lapse images were recorded at 2-s intervals for 1 min before and 1–2 min after the induction of $Ca^{2+}$ transients. Imaging data were collected using identical imaging and processing parameters for a given experimental set. All experiments were repeated at least 3 times, and the representative data were shown.

### Calcium measurements

Basal or treatment-induced live-cell $Ca^{2+}$ measurements were carried out using genetically encoded GCaMP-based probes[69]. In order to measure changes in $Ca^{2+}$-signaling in different subcellular locations, neurons were transfected at DIV 2–3 with organelles-specific $Ca^{2+}$ sensors targeted either to mitochondria (mito GCepia-3 or 2), endoplasmatic reticulum (ER-GCamp6-210) or cytosol (jGCaMP7b-WPRE). Ratiometric $Ca^{2+}$ imaging was performed 6–7 days later by recording GCaMP's green fluorescence emission from 495 to 555 nm following both 405 and 488 nm laser excitation. The time-lapse $Ca^{2+}$ imaging experiments with glutamate (100 μM glutamate and 10 μM glycine) or DHPG treatment were performed at DIV 9–10.

For experiments involving outer mitochondrial membrane permeabilization, the transfected neurons were incubated at DIV 9–10 in Krebs-Ringer solution containing 1 mM $Ca^{2+}$ and 5 μM Mag-Fluo-4 AM for 1 hour at 37 °C. Then cells were permeabilized in a basic solution containing 10 mM EGTA, 30 mM N, N-bis[2-hydroxyethyl]−2-aminoethanesulfonic acid (BES, pH 7.1), 1 mM free $Mg^{2+}$, 20 mM taurine, 5.56 mM glutamic acid, 1.5 mM malic acid, 3.9 mM $K_2HPO_4$, 0.5 mM dithiothreitol and 3.16 mM NaATP; ionic strength was adjusted to 160 mM with potassium methanesulfonate (pH 7.1) with saponin (50 μg/ml) for 16 minutes at 4 °C and washed with the same solution containing additionally sodium azide (3 mM). Images were collected using an LSM 780 confocal microscope (Plan-Apochromat ×20/0.5 objective) at room temperature. From each dish, one random field with size 424 × 424 μm was selected and imaged at 3-second intervals over 10 minutes. After 20 frames, the $CaCl_2$ was added to achieve pCa

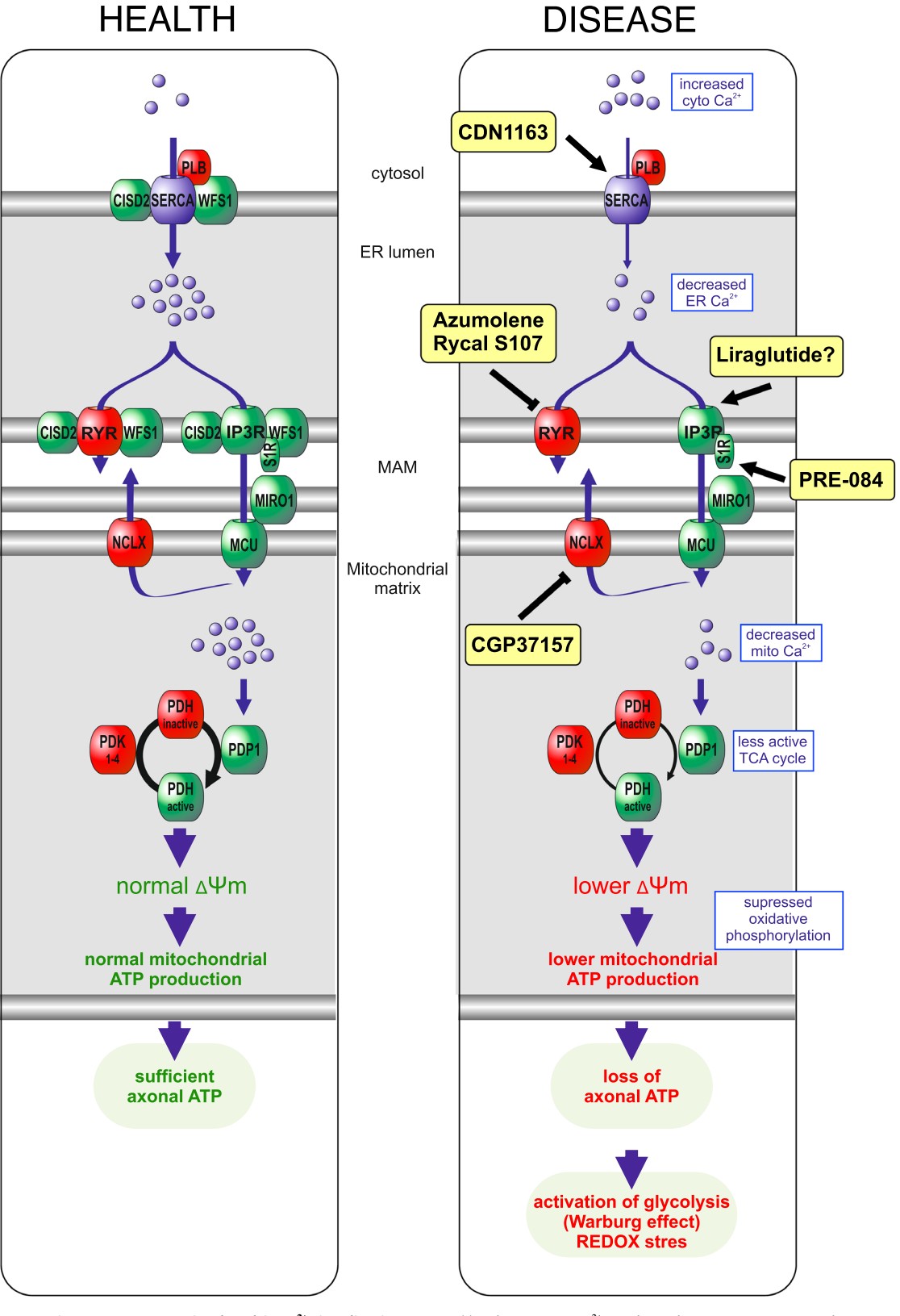

**Fig. 9 | The main strategies to restore ER-mitochondria Ca²⁺ signaling in WS neurons and to ensure the normal function of numerous Ca²⁺-dependent mitochondrial processes.** Loss of ER Ca²⁺ content and ER-mitochondrial contact sites in WFS1- or CISD2-deficient neurons results in diminished IP3R-mediated Ca²⁺ transfer from ER to mitochondria. Reduced mitochondrial Ca²⁺ levels suppress the TCA cycle and mitochondrial ATP production, leading to the bioenergetic deficit and redox stress, particularly at axonal terminals. Potential interventions include: (1) enhancing ER Ca²⁺ uptake with SERCA activators such as CDN1163, (2) suppressing ER Ca²⁺ release or leakage via ryanodine receptors with Rycal S107 or azumolene, (3) activating IP3R-mediated Ca²⁺ flux with sigma-1 receptor (S1R) agonists like PRE-084, (4) inhibiting Ca²⁺ extrusion from the mitochondrial matrix by targeting the Na⁺/Ca²⁺ exchanger with CGP37157 and (5) potentially improving Ca²⁺ homeostasis with the GLP-1 agonist liraglutide, although its mechanisms in this context remain unclear.

6.5. Change of MagFluo-4 AM signal from non-transfected or transfected individual neurons was measured.

For thapsigargin-induced $Ca^{2+}$ release experiments, primary cortical neurons were transfected at DIV 2–3 with either scrambled, *Wfs1*, or *Cisd2* shRNA along with a plasmid containing the cytosolic $Ca^{2+}$ sensor jGCaMP7b-WPRE. At DIV 9–10, the growth media were replaced with Krebs-Ringer modified solution lacking $Ca^{2+}$ (135 mM NaCl, 5 mM KCl, 1 mM $MgSO_4$, 0.4 mM $KHCO_3$, 20 mM HEPES, Glucose 5.5 mM plus 0.5 mM EGTA; pH adjusted to 7.4), and ratiometric $Ca^{2+}$ imaging was conducted by recording the green fluorescence emission of GCaMP from 495 to 555 nm following excitation by both 405 nm and 488 nm lasers. Images were acquired on a Zeiss LSM 980 (Zeiss, Germany) with a 20×/0.8 objective at 37 °C. Cells were treated with 2 µM Thapsigargin in a $Ca^{2+}$-free solution, followed by the addition of a $Ca^{2+}$-containing solution to achieve a final concentration of 10 to 20 mM free $Ca^{2+}$ in the imaging media.

## ATP/ADP ratio
Relative intracellular ATP/ADP ratio was estimated using the genetically encoded fluorescent ratiometric probe PercevalHR[70], which was previously validated for our experimental settings. Neurons (specifically, depending on the experiment, either axonal shaft or endings) expressing the ATP/ADP ratio sensor PercevalHR were excited using 405 nm and 488 laser lines and emission was collected using a 495–555 nm emission window. Data are expressed as the ratio of fluorescence emission from axons evoked by 488 nm excitation divided by 405 nm excitation (F488 nm/F405 nm).

## NADH/NAD$^+$ ratio
For live cell imaging of cytosolic NADH/NAD$^+$ redox state, neurons were transfected either with Peredox-mCitrine or SoNar biosensors and plasmids of interest on DIV 2–3 and imaged 6–7 days later using confocal microscope. Both reporters contain subunits of Rex, a bacterial NADH-binding protein from Thermus aquaticus (T-Rex) as detector domain and are designed pH insensitive. SoNar is an intrinsically ratiometric sensor involving circularly permuted yellow protein (cpYFP) and having opposing fluorescence responses upon NAD$^+$ or NADH-binding when excited at 420 and 500 nm[36], whereas Peredox includes green T-Sapphire (cpFP with excitation peak around 400 nm and an emission peak ~510 nm) in sensor domain and is made to be ratiometric by tagging with a yellow fluorescent mCitrine protein for normalization[35]. Due to the better basal fluorescence intensity over SoNar in primary cortical neurons, the Peredox-mCitrine reporter was preferred in majority of our experimental settings. For imaging, the NADH/NAD$^+$ metabolic state, neurons expressing Peredox-mCitrine were excited using a 405 nm laser line and emission was collected using a 490–600 nm emission window. In order to normalize the NADH signal between cells for different protein expression, the mCitrine was excited at 488 nm and emission was measured in 500–600 nm range. NADH/NAD$^+$ ratio data was presented as the ratio of fluorescence emission from axons or axonal endings evoked by 405 nm excitation divided by 488 nm excitation (F405 nm/F488 nm).

## Quantification of MAMs
Cortical neurons plated on a 96-well black/clear bottom plate (Thermo Fisher) were transfected with pMAMtracker-Luc, piRFP720 and plasmids of interest at DIV 2–3. Five days post-transfection the media was replaced with 50 µl of Nano-Glo® Live Cell Reagent in DPBS and the luminescence and fluorescence were recorded using the GloMax Multi Detection System (Promega) and Odyssey®M imaging system (Li-Cor), respectively. Luminescence for the complemented NanoBiT was normalized to cytosolic iRFP720 fluorescence.

## Mitochondrial membrane potential
Primary neurons were transfected with validated siRNA against Wfs1[11] and Cisd2 (*Cisd2* siRNA suppressed 52 ± 10% of endogenous CISD2 expression in primary cortical cells according to RT-PCR, $n = 3$) using the N-TER Nanoparticle siRNA Transfection System (Sigma-Aldrich). At first, a combination of target and scrambled siRNA (20 nM) diluted in siRNA buffer and N-TER transfection reagent diluted in ddH20 was preincubated at RT for 20 min. The growth medium was then replaced with Opti-MEM I containing the target or scrambled siRNA mixture. After three hours at 37 °C, Opti-MEM I was changed to Neurobasal-A medium containing B-27 supplement, 2 mM GlutaMAX-I, and 100 µg/ml gentamicin. Finally, neurons were incubated for 48–72 h in a humidified 5% $CO_2$/95% air incubator at 37 °C.

Next, siRNA-transfected dishes were loaded with 10 µM JC-10 (Enzo Life Sciences, ENZ-52305) dissolved in culture media and incubated at 37 °C for 25 min. Dye-loaded neurons were kept in Krebs-Ringer solution supplemented with 1 mM $Ca^{2+}$ and visualized using a laser scanning confocal microscope equipped with a LCI Plan-Neofluar ×63/1.3 water immersion DIC M27 objective. In the end, to obtain background values, dishes were treated with 10 µM FCCP.

## Mitochondrial density and length
To measure mitochondrial length and density in axons, the neuronal cultures were transfected with EGFP, mitochondrial pDsRed2, scrambled shRNA and indicated shRNA under vehicle or treatment conditions and imaged 6–7 days after transfection. Fluorescence images of 10 randomly selected axons from each dish were taken with an Olympus IX70 inverted microscope equipped with WLSM PlanApo ×40/0.90 water immersion objective and Olympus DP70 CCD camera. Morphometric analysis was done using MicroImage software (Media Cybernetics, Bethesda, MD). For mitochondrial length and density measurements, at least 40 axons per group were analyzed.

## Mitophagy
Neuronal cultures transfected with mitochondrially targeted Keima and scrambled shRNA or targeted shRNA under vehicle or treatment conditions were examined 6–7 days after transfection. The excitation spectrum of mtKeima shifts from 440 to 586 nm when mitochondria are delivered to acidic lysosomes, allowing quantification of mitophagy. Images of 10 randomly selected neuronal bodies per dish were acquired by a laser scanning confocal microscope (LSM 510 Duo) using the laser lines 458 nm (green, mitochondria at neutral pH) and 561 nm (red, mitochondria under acidic pH), and red dots were counted blindly using LSM5 software.

## Axonal growth
For the measurement of the main branch of the axon, cortical neurons were transfected at DIV1 with neuron-specific pAAV-hSyn-DsRedExpress and scrambled shRNA or targeted shRNA and treated with a vehicle or a chemical of interest. Images of randomly selected cortical neurons (containing minimum 40 axons per group) were captured 2 days after transfection using a fluorescence microscope (Olympus IX70, ×20/0.70 water immersion objective). Neuronal reconstructions were established using the Fiji plug-in NeuronJ and the length of the main branch of the axon was measured using Fiji.

## Immunoprecipitation
Co-immunoprecipitation from non-transfected HEK293 cells was performed as described in the Supplementary Methods using rec-Protein G-Sepharose 4B beads. Note the modification of the elution protocol for co-precipitated proteins. Co-immunoprecipitation of EGFP-WFS1 and CISD2-myc from transfected HEK293 cells was performed according to the GFP-Trap manufacturer's (Proteintech) protocol as detailed in the Supplementary Methods. For co-immunoprecipitation of RyR-interacting proteins, whole brain lysates from six-month-old

mice were obtained, and co-immunoprecipitation was performed using Protein G Dynabeads as described in the Supplementary Methods.

## Statistics and reproducibility

Data are presented as the mean ± SEM. The D'Agostino-Pearson omnibus test was used to assess the normality of distribution. The equality of variances was tested using the $F$ test for two conditions or the Brown and Forsythe test for more than two conditions. The outliers were removed in the case of $Ca^{2+}$ experiments (to exclude bursting neurons), axonal growth experiments (to exclude neurons with retarded growth), and MAM experiments (to exclude wells with too low or high number of cells) using the ROUT method ($Q = 1\%$).

Two-tailed Student t-tests, Ordinary one-way ANOVA followed by Sidak's multiple comparison test, or Brown-Forsythe ANOVA followed by Dunnett's T3 multiple comparisons test were used to compare the parametric data. Two-tailed Mann–Whitney $U$ test or Kruskal–Wallis tests followed by the Dunn test were used to analyze the non-parametric data. Two-way ANOVA was used to obtain the interaction between the treatments. $P$ values of <0.05 were considered statistically significant.

Curves depicted in Figs.1a, b, and 3b were repeated independently three times. Colocalization experiments depicted in Fig. 7a, b were independently performed three times. Immunoprecipitation blots depicted in Fig. 7d, e were independently performed three times, and those in Fig. 7c, f were performed twice.

## Reporting summary

Further information on research design is available in the Nature Portfolio Reporting Summary linked to this article.

## Data availability

CISD2 crystal structure used in this article is available in the Protein Data Bank under the following accession code: 3FNV. Source data are provided with this paper.

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

## Acknowledgements

We thank Dr. G. Szabadkai, Dr. K. Yamanaka, Dr. Y. Yang, Dr. D. Yule, and Addgene, as well as the colleagues depositing their plasmids there, for providing the plasmids used in this study. We thank Ulla Peterson for her preparation of primary neurons. This work was supported by grants from the Estonian Research Council (PRG400 and PRG2145). G.B. and A.K. were supported by the Research Foundation – Flanders (FWO)

(G081821N) and the Central European Leuven Strategic Alliance (CELSA/18/040). A.K. was supported by the Chan Zuckerberg Initiative. G.B. was further supported the Stichting Alzheimer Onderzoek (SAO IP3 RECEPTOR to G.B.) and Eye Hope Foundation/King Baudouin Foundation (grantbuitenoproep_FondsEyeHope_J1160630_Bultynck). A.K. and V.V. were supported by France-Estonia Partenariats Hubert Curien (Parrot 42402TF). V.V. was supported by MILEAGE Consortium Agreement Horizon 2020 n°734931. M.C. was supported by the Slovak Research and Development Agency (APVV-21-0473) and Building-up Centre for Advanced Materials application of the Slovak Academy of Sciences supported by the Research & Innovation Operational Programme funded by the ERDF (ITMS project code 313021T081). T.V. was supported by a FWO fellowship 12ZG121N.

## Author contributions

M.L., A.V., E.V., J.L., T.V., T.-F.T., G.B., V.V., and A.K. participated in the conceptualization of the project. M.L., A.V., G.B., V.V., and A.K. conceived and designed the project. M.L., A.V., D.S., V.C., R.G., M.K., L.J., Z.H., and A.K. designed the experiments. M.L., A.V., M.K., R.G., L.J., Z.H., M.C., A.Z., V.C., M.A.H., D.S., Y.-L.H., N.G., and M.M. performed and analyzed experiments. E.V., J.L., T.V., T.-F.T., M.P., and G.B. provided reagents. M.L., A.V., V.V., and A.K. wrote the original draft. All authors reviewed and edited the manuscript.

## Competing interests

The authors declare no competing interests.
