## [Peer Review File · Nature Communications]

ER calcium depletion as a key driver for impaired
ER-to-mitochondria calcium transfer and mitochondrial
dysfunction in Wolfram syndromeREVIEWER COMMENTS

Reviewer #1 (Remarks to the Author):

The study delves into the intricacies of Wolfram Syndrome (WS), particularly focusing on the interactions between ER and mitochondrial calcium (Ca^{2+}) homeostasis and the implications of Wfs1 and CISD2 proteins. Notable findings suggest compromised ER-to-mitochondria Ca^{2+} transfer, attributed to decreased SERCA activity and Ca^{2+} leakage via RYR receptors, which subsequently hinders mitochondrial ATP synthesis. An intriguing observation is the physical interaction between Wfs1 and CISD2, suggesting a compensatory relationship, thereby hinting at potential therapeutic avenues.

Despite the efforts, there are several areas of concern:

1. The roles of Wfs1 and CISD2 in ER calcium homeostasis and their influence on mitochondrial calcium flow have been previously explored, albeit with some lingering mechanistic uncertainties. Reports highlighting the phenotypic overlap of their loss of function and their closeness in organellar biology already exist. Hence, the mitochondrial dysfunction resulting from Wfs1 and CISD2 loss-of-function is somewhat anticipated. The novelty of this research needs some justification.

2. There are unclear aspects of the molecular mechanisms:

- The effects of Wfs1 or CISD2 knock-down on SERCA and RyR activity require further mechanistic elucidation.
- The physiological significance and implications of the Wfs1-CISD2 interaction remain ambiguous (Figure 6).
- The study needs to clarify how a deficiency in Wfs1 or CISD2 can be offset by overexpression of the other mechanistically (Figure 7).

3. The research centers on the implications of Wfs1 or CISD2 knock-down on steady ER and axoplasm Ca^{2+} levels, emphasizing SERCA and RyR2 functions. The current experimental setup might not robustly support the results, especially concerning extracellular Ca^{2+} reuptake. An effective approach might entail measuring Ca^{2+} concentrations in the ER and axoplasm after targeted Ca^{2+} release in a Ca^{2+} -free buffer, subsequently reintroducing external calcium during channel activation.

4. To further validate the knock-down effect on channel functionality, it would be insightful to directly monitor variations in ER Ca²⁺ absorption during SERCA augmentation or inhibition, as well as Ca²⁺ release during RyR2 inhibition, preferably in a Ca²⁺-free environment. The authors should check if SERCA, RyR2, and IP3Rs expression levels under Wfs1 or CISD2 knock-down conditions.

5. Several sections, including Figure 3, seem to attribute the mitochondrial Ca²⁺ up- or downregulation seen in this study to Ca²⁺ transfer through MAM, yet without definitive experimental evidence. IP3R localization is not restricted to the MAM, further experimental clarification is necessary.

6. Given the variety of Wfs1 and CISD2 mutations associated with WS, exploring the correlation between disease severity and mutation impacts on ER and mitochondrial calcium dynamics would be informative. Pairing Wfs1/CISD2 knock-down with WS-related mutants could provide further insights into the link between this study's core observation and WS.

7. Figure 6 offers potentially interesting data but lacks depth. All tests were conducted under overexpression conditions. The rationale for utilizing the mitochondrial marker in (B) is ambiguous. The physiological and disease relevance of the potential interaction remains unexamined.

Reviewer #2 (Remarks to the Author):

Wolfram syndrome is a rare neurodegenerative disease characterized by a diabetes insipidus, a diabetes mellitus, an optic atrophy and a deafness. More symptoms may be present such as ataxia, kidney issues, anxiety and suicide. Unfortunately, patients die around 35 years old following brainstem atrophy. As of today no treatment is available and there is an urgent need to at least improve the quality of life of the patients. Wolframin, the protein encoded by WFS1 gene, is a transmembrane endoplasmic reticulum resident protein, has been shown to regulate the unfolded protein response triggered by an endoplasmic reticulum stress. Recently, another physiopathological signaling pathway emerged. Wolframin is supposed to play also a role in the communication between endoplasmic reticulum and mitochondria at the mitochondria-associated ER membranes or MAMS. Indeed, wolframin appears to be a regulator of calcium homeostasis of the MAMS. Two types of wolfram syndrome are described, the type 1, involving WFS1 and the type 2 (less frequent) involving CISD2.

The work presented by Mailis Liiv et al. aims at deciphering how WFS1 and CISD2 loss of function impairs the endoplasmic reticulum to mitochondria calcium transfer and the associated

mitochondrial function. Using state of the art calcium imaging in primary neuronal culture, they elegantly demonstrated that in fact wolframin interacts with CISD2 to regulate the axonal endoplasmic reticulum calcium content. This leads to a decrease in the IP3R mediated calcium transfer to the mitochondria, reducing mitochondrial calcium content. Thus, this induced an increased axoplasmic NADH/NAD⁺ ratio. Finally, they tested different small active molecule to restore calcium homeostasis that may be relevant to treat Wolfram syndrome patients.

This topic is of highly importance since it brings significant insight in the main physiopathological mechanism leading to Wolfram syndrome and propose some therapeutic intervention to slow down or block the progression of this devastating pathology. Overall, the experiments are well conducted and the results are of importance since they were conducted in the principal affected cell type, the neurons. The conclusion are supported by the results.

Nevertheless, I have some questions:

- 1) Are WFS1 and CISD2 expressed in the presynaptic compartment? If yes, does their knock down in this compartment impact ER homeostasis in a similar way?
- 2) Does the MAMs are affected? Contact between endoplasmic reticulum and mitochondria are effective in the axon too.

Minor comments

Page 3, line 43, wolframin should be replaced by WFS1.

P3, line 62, I think that the correct number 13 reference should be Zatyka M et al., Stem Cell Reports. 2023 May 9; 18(5):1090-1106 rather than Zatyka et al, Hum Mol Genet.

RESPONSE TO REFEREES

Reviewer #1 (Remarks to the Author):

The study delves into the intricacies of Wolfram Syndrome (WS), particularly focusing on the interactions between ER and mitochondrial calcium (Ca^{2+}) homeostasis and the implications of Wfs1 and CISD2 proteins. Notable findings suggest compromised ER-to-mitochondria Ca^{2+} transfer, attributed to decreased SERCA activity and Ca^{2+} leakage via RYR receptors, which subsequently hinders mitochondrial ATP synthesis. An intriguing observation is the physical interaction between Wfs1 and CISD2, suggesting a compensatory relationship, thereby hinting at potential therapeutic avenues.

We thank Reviewer 1 for the insightful recommendations, which have greatly expanded the depth of our study.

Despite the efforts, there are several areas of concern:

1. The roles of Wfs1 and CISD2 in ER calcium homeostasis and their influence on mitochondrial calcium flow have been previously explored, albeit with some lingering mechanistic uncertainties. Reports highlighting the phenotypic overlap of their loss of function and their closeness in organellar biology already exist. Hence, the mitochondrial dysfunction resulting from Wfs1 and CISD2 loss-of-function is somewhat anticipated. The novelty of this research needs some justification.

We agree with the Reviewer that several papers have explored ER and mitochondrial calcium homeostasis alterations. At the same time, no research paper has thoroughly investigated calcium levels in the ER, mitochondrial, and cytosol compartments, and, importantly, their interactions, albeit in axons. By using this systematic approach, we can better understand how calcium levels change within different cellular compartments in Wolfram Syndrome, helping to clarify the underlying mechanisms. We deliberately avoided a simplistic descriptive approach, opting to delve deeply into understanding how alterations in one compartment impact another.

Moreover, our study reveals an interaction between WFS1 and CISD2, as well as their co-localization within protein complexes alongside key ER calcium regulatory proteins. This allows us to suggest that WFS1 and CISD2 function as ER-associated regulatory proteins transmitting signals from the endoplasmic reticulum stress and redox signaling pathways to the ER calcium handling apparatus (see also reply to next comment) and further to the mitochondrial matrix.

Furthermore, although it is known that ER dysfunction affects mitochondrial calcium flux, it was not known how this leads to a bioenergetic deficit. We propose a plausible explanation how mitochondrial dysfunctions could be corrected by manipulating calcium pump and channels.

We have tested various compounds, including azumolene (a non-fluorescent analogue of dantrolene that have been in WS clinical trials) and liraglutide (used off-label for WS patients), revealing potential correction strategies viable for therapeutic intervention.

We revised the initial paragraph of the Discussion section (lines 298-307 in the revised manuscript) to emphasize these key points more clearly. Furthermore, the new experiments conducted to address

the remaining comments from the Reviewers have substantially enhanced the depth of our conclusions.

2. There are unclear aspects of the molecular mechanisms:

- The effects of Wfs1 or CISD2 knock-down on SERCA and RyR activity require further mechanistic elucidation.
- The physiological significance and implications of the Wfs1-CISD2 interaction remain ambiguous (Figure 6).
- The study needs to clarify how a deficiency in Wfs1 or CISD2 can be offset by overexpression of the other mechanistically (Figure 7).

We acknowledge the reviewers' insightful feedback, recognizing the need for further clarification in the manuscript.

In the revised version, we demonstrate that endogenous WFS1 strongly co-immunoprecipitates with pulled-down CISD2 and that both endogenous WFS1 and CISD2 co-immunoprecipitates with SERCA2 and RyR2 (Fig. 7d-f and lines 254-258). These data indicate that the interaction is relevant under physiological conditions, not only in artificial overexpressing systems.

As a matter of fact, both WFS1 and CISD2 are molecular partners of SERCA2 and RYR. Considering their notable structural differences, it is plausible to propose that these proteins mediate distinct cellular signals. WFS1 is implicated as one of the downstream components of the unfolded protein response (Fonseca et al. 2005 and 2010). On the other hand, CISD2, featuring an iron-sulfur cluster, appears to be involved in responding to redox stress (Li et al. 2017, Conlan et al. 2009). Consequently, it is reasonable to speculate that Wfs1 and CISD2, acting as sensors for ER stress and redox stress, respectively, synergize to integrate these converging signals, thereby regulating cellular ER calcium handling.

In this scenario, it is reasonable to expect that the loss of one pathway, whether it be WFS1 or CISD2, could potentially be mitigated through the overactivation of an alternative pathway, as demonstrated in Figure 8 (Fig. 7 in earlier version). We also integrated this explanation into the Discussion section of the revised manuscript (lines 387-401).

3. The research centers on the implications of Wfs1 or CISD2 knock-down on steady ER and axoplasm Ca²⁺ levels, emphasizing SERCA and RyR2 functions. The current experimental setup might not robustly support the results, especially concerning extracellular Ca²⁺ reuptake. An effective approach might entail measuring Ca²⁺ concentrations in the ER and axoplasm after targeted Ca²⁺ release in a Ca²⁺-free buffer, subsequently reintroducing external calcium during channel activation.

We thank the Reviewer for the important comment. We performed the suggested experiments and measured the thapsigargin-induced surge of cytosolic calcium in the calcium-free buffer. The results show that in Wfs1 and Cisd2 deficient neurons, there is significantly less RyR-releasable calcium in the ER, even in a calcium-free medium (Fig. 2a). This excludes the role of extracellular calcium reuptake and confirms that the primary defect should be related to the decreased ER calcium uptake or/and increased prior calcium leak. Moreover, our experiments in plasma membrane-permeabilized neurons, where the influence of external calcium reuptake was removed, also showed lower ER calcium uptake in Wfs1 and Cisd2 deficient neurons. To emphasize this point, we transferred these data from the Supplementary to the main figure (Fig. 1j). We also rewrote the respective paragraphs in the Results section (lines 108-111 and 119-127) to emphasize that the primary intrinsic deficit lies in ER Ca²⁺ handling and is not significantly influenced by external factors. *Note that thapsigargin-*

induced calcium release is relatively small in primary neurons when compared to traditionally used cell lines and is difficult to use for routine measurements.

4. To further validate the knock-down effect on channel functionality, it would be insightful to directly monitor variations in ER Ca²⁺ absorption during SERCA augmentation or inhibition, as well as Ca²⁺ release during RyR2 inhibition, preferably in a Ca²⁺-free environment. The authors should check if SERCA, RyR2, and IP3Rs expression levels under Wfs1 or Cisd2 knock-down conditions.

In response to reviewer feedback, we conducted additional experiments to investigate the impact of SERCA enhancement or inhibition on calcium uptake in permeabilized neurons, where the influence of external calcium is eliminated. Our findings reveal that both enhancing SERCA (via overexpression or activation) and inhibiting it significantly increase or decrease ER calcium uptake.

We also assessed calcium release during RyR2 inhibition in response to thapsigargin in intact neurons in a calcium-free environment. The RYR inhibitors effectively reduce RyR-mediated calcium release in these settings. Both sets of data are now presented in Supplementary Figure 2.

We conducted a single-cell (neuron) expression analysis, indicating that while RyR2 and IP3R expression levels remained unaffected under these conditions, the levels of SERCA2 were lower in the Wfs1 or Cisd2 knock-down neurons. These results are included in Supplementary Figure 3 in the revised manuscript. This decrease in SERCA levels may be specific to neurons, as we did not observe a significant decline in SERCA2 levels in WFS1 knock-down in the PC6 cell line. That specificity might also explain the discrepancy with earlier data about SERCA expression observed in other cell lines (Zatyka et al. 2015).

5. Several sections, including Figure 3, seem to attribute the mitochondrial Ca²⁺ up- or downregulation seen in this study to Ca²⁺ transfer through MAM, yet without definitive experimental evidence. IP3R localization is not restricted to the MAM, further experimental clarification is necessary.

We understand the importance of providing clear experimental clarification for this question. In our revised version, we have addressed this concern by utilizing a new-generation, reversible MAM probe called MAM-tracker (Figure 4A). Our results, presented in Figure 4B, demonstrate a significant reduction in MAM levels in neurons deficient in WFS1 and Cisd2.

Additionally, the overexpression of GRP75, known to link the mitochondrial VDAC and ER-located IP3R, restored the MAM levels as well as mitochondrial calcium uptake in the Wfs1 deficient neurons (Fig. 4 C and D). We believe that this experiment serves as definitive proof that the diminished calcium transfer through MAMs contributes to the decreased mitochondrial calcium uptake in these neurons. We have also significantly modified the Results and Discussion sections (lines 188-204 and 347-353) to highlight these results.

6. Given the variety of Wfs1 and Cisd2 mutations associated with WS, exploring the correlation between disease severity and mutation impacts on ER and mitochondrial calcium dynamics would be informative. Pairing Wfs1/Cisd2 knock-down with WS-related mutants could provide further insights into the link between this study's core observation and WS.

We thank the Reviewer for the valuable suggestion. As suggested, we tested the efficacy of wild-type WFS1 as well as four different disease-linked mutations to rescue mitochondrial Ca²⁺ homeostasis in WFS1 knock-down neurons. The mutations A684V and P724L are associated with a moderately severe disease phenotype, whereas the mutations K836N and E864K, as outlined in an in-depth review (De Franco et al. 2017), result in less severe manifestations of the disease. Our findings demonstrate that

wild-type, K836N, and E864K WFS1 were able to fully rescue the effects of WFS1 knock-down on mitochondrial calcium levels. However, the A684V and P724L mutations had no rescue capacity. These results suggest that mutations associated with varying degrees of disease severity also affect mitochondrial calcium dynamics to different extents.

We included the results of these experiments as Supplementary Figure 5 in the revised manuscript and also introduced an additional paragraph into the Results section (lines 163-172).

7. Figure 6 offers potentially interesting data but lacks depth. All tests were conducted under overexpression conditions. The rationale for utilizing the mitochondrial marker in (B) is ambiguous. The physiological and disease relevance of the potential interaction remains unexamined.

We conducted additional experiments to confirm the interaction between endogenous WFS1 and CISD2. As shown in Figure 7d, endogenous WFS1 strongly co-immunoprecipitates with CISD2 in HEK cells. Furthermore, we replicated this immunoprecipitation assay in PC6 cells, yielding consistent results. Additionally, we demonstrate that both endogenous WFS1 and CISD2 co-immunoprecipitate with SERCA2 (Fig. 7f) and RyR2 (previously shown in Fig. 2a, now included in Fig. 7e; lines 254-258). These additional findings provide further evidence that the interaction occurs between endogenous proteins.

We also included the super-resolution images of WFS1-EGFP, and CISD2-YPet, revealing that both are present in axon terminals as suggested by Reviewer 2 (Fig. 7a). We also moved the panel showing the proximity ligation assay to Supplementary Figure 7 and also excluded the subpanel with the mitochondrial marker, as suggested.

While we acknowledge that proposing the precise molecular mechanism underlying the disease relevance of this interaction is not within the scope of our study, we suggest that WFS1 and CISD2 may act together, or at least in concert, to regulate SERCA and possibly other ER calcium-handling proteins. They are likely to participate in parallel pathways, such as ER- and redox stress, which converge at the level of SERCA, and presumably also RyR and IP3R, rather than functioning solely as upstream or downstream partners of one pathway.

Reviewer #2 (Remarks to the Author):

Wolfram syndrome is a rare neurodegenerative disease characterized by a diabetes insipidus, a diabetes mellitus, an optic atrophy and a deafness. More symptoms may be present such as ataxia, kidney issues, anxiety and suicide. Unfortunately, patients die around 35 years old following brainstem atrophy. As of today no treatment is available and there is an urgent need to at least improve the quality of life of the patients. Wolframin, the protein encoded by WFS1 gene, is a transmembrane endoplasmic reticulum resident protein, has been shown to regulate the unfolded protein response triggered by an endoplasmic reticulum stress. Recently, another physiopathological signaling pathway emerged. Wolframin is supposed to play also a role in the communication between endoplasmic reticulum and mitochondria at the mitochondria-associated ER membranes or MAMs. Indeed, wolframin appears to be a regulator of calcium homeostasis of the MAMs. Two types of wolfram syndrome are described, the type 1, involving WFS1 and the type 2 (less frequent) involving CISD2.

The work presented by Mailis Liiv et al. aims at deciphering how WFS1 and CISFD2 loss of

function impairs the endoplasmic reticulum to mitochondria calcium transfer and the associated mitochondrial function. Using state of the art calcium imaging in primary neuronal culture, they elegantly demonstrated that in fact wolframin interacts with CISD2 to regulate the axonal endoplasmic reticulum calcium content. This leads to a decrease in the IP3R mediated calcium transfer to the mitochondria, reducing mitochondrial calcium content. Thus, this induced an increased axoplasmic NADH/NAD⁺ ratio. Finally, they tested different small active molecule to restore calcium homeostasis that may be relevant to treat Wolfram syndrome patients.

This topic is of highly importance since it brings significant insight in the main physiopathological mechanism leading to Wolfram syndrome and propose some therapeutic intervention to slow down or block the progression of this devastating pathology. Overall, the experiments are well conducted and the results are of importance since they were conducted in the principal affected cell type, the neurons. The conclusion are supported by the results.

We appreciate Reviewer 2's insightful questions, which have prompted us to delve deeper into the involvement of mitochondrial-associated membranes (MAMs) in WS.

Nevertheless, I have some questions:

1) Are WFS1 and CISD2 expressed in the presynaptic compartment? If yes, does their knock down in this compartment impact ER homeostasis in a similar way?

Super-resolution images of WFS1-EGFP and CISD2-YPet reveal that both are present in axon terminals (Fig. 7a in the revised version). While not all axonal terminals are yet presynaptic at DIV 10 when our experiments were conducted, this observation strongly suggests that WFS1 and CISD2 are also present in presynaptic axonal terminals in mature neurons.

Although it is not possible to specifically target an ER calcium probe to the ER of axonal terminals, our direct measurement of cytosolic calcium in axonal terminals and ER calcium in close proximity allows us to conclude that knock-down affects the ER similarly in this region.

2) Does the MAMs are affected? Contact between endoplasmic reticulum and mitochondria are effective in the axon too.

We conducted further experiments to address this comment, employing a new-generation reversible MAM probe named MAM-tracker (Fig. 4A). Our results indicate an apparent reduction in MAM levels in neurons deficient in WFS1 and CISD2 (Fig 4B and C).

Additionally, the overexpression of GRP75, known to link the mitochondrial VDAC and ER-located IP3R, restored the MAM levels as well as mitochondrial calcium uptake in the Wfs1 deficient neurons (Fig. 4 C and D). We believe that this experiment serves as definitive proof that the diminished calcium transfer through MAMs contributes to the decreased mitochondrial calcium uptake in these neurons. We also modified the Results and Discussion sections accordingly (lines 188-204 and 347-353).

Minor comments

Page 3, line 43, wolframin should be replaced by WFS1.

We appreciate the reviewer's attention. While we acknowledge that "WFS1" is more widely used in the literature to refer to the protein, it's important to note that according to UniProt, "wolframin" remains the official name of the protein. Therefore, we believe it is more accurate to use the official

name "wolframin" at the beginning of our Introduction and then transition to "WFS1" in subsequent text.

P3, line 62, I think that the correct number 13 reference should be Zatyka M et al., Stem Cell Reports. 2023 May 9; 18(5):1090-1106 rather than Zatyka et al, Hum Mol Genet.

We corrected the reference.

REVIEWERS' COMMENTS

Reviewer #1 (Remarks to the Author):

My concerns were addressed adequately. I do not have further questions.

Reviewer #2 (Remarks to the Author):

The authors answered to all my criticisms.